# Cross-protection against African swine fever virus upon intranasal vaccination is associated with an adaptive-innate immune crosstalk

Laia Bosch-Camós[1,2☯], Uxía Alonso[1,2☯], Anna Esteve-Codina[3,4], Chia-Yu Chang[1,2], Beatriz Martín-Mur[3], Francesc Accensi[1,5], Marta Muñoz[1,2], María J. Navas[1,2], Marc Dabad[3], Enric Vidal[1,2], Sonia Pina-Pedrero[1,2], Patricia Pleguezuelos[1,2], Ginevra Caratù[3], María L. Salas[6], Lihong Liu[7], Stanimira Bataklieva[8], Boris Gavrilov[8], Fernando Rodríguez[1,2*], Jordi Argilaguet[1,2*]

**1** Unitat mixta d'Investigació IRTA-UAB en Sanitat Animal. Centre de Recerca en Sanitat Animal (CReSA), Campus de la Universitat Autònoma de Barcelona (UAB), Bellaterra, Spain, **2** IRTA. Programa de Sanitat Animal. Centre de Recerca en Sanitat Animal (CReSA), Campus de la Universitat Autònoma de Barcelona (UAB), Bellaterra, Spain, **3** CNAG-CRG, Centre for Genomic Regulation (CRG), Barcelona Institute of Science and Technology (BIST), Barcelona, Spain, **4** Universitat Pompeu Fabra (UPF), Barcelona, Spain, **5** Departament de Sanitat i Anatomia animals. Facultat de Veterinària, Campus de la Universitat Autònoma de Barcelona (UAB), Bellaterra, Spain, **6** Centro de Biología Molecular Severo Ochoa, Consejo Superior de Investigaciones Científicas and Universidad Autònoma de Madrid, Madrid, Spain, **7** National Veterinary Institute (SVA), Uppsala, Sweden, **8** Biologics Development, Huvepharma, 3A Nikolay Haytov Street, Sofia, Bulgaria

☯ These authors contributed equally to this work.
* fernando.rodriguez@irta.cat (FR); jordi.argilaguet@irta.cat (JA)

**Data Availability Statement:** All raw and processed sequencing data generated in this study have been submitted to the NCBI Gene Expression

## Abstract

African swine fever virus (ASFV) is causing a worldwide pandemic affecting the porcine industry and leading to important global economic consequences. The virus causes a highly lethal hemorrhagic disease in wild boars and domestic pigs. Lack of effective vaccines hampers the control of virus spread, thus increasing the pressure on the scientific community for urgent solutions. However, knowledge on the immune components associated with protection is very limited. Here we characterized the *in vitro* recall response induced by immune cells from pigs intranasally vaccinated with the BA71ΔCD2 deletion mutant virus. Vaccination conferred dose-dependent cross-protection associated with both ASFV-specific antibodies and IFNγ-secreting cells. Importantly, bulk and single-cell transcriptomics of blood and lymph node cells from vaccinated pigs revealed a positive feedback from adaptive to innate immunity. Indeed, activation of Th1 and cytotoxic T cells was concomitant with a rapid IFNγ-dependent triggering of an inflammatory response characterized by TNF-producing macrophages, as well as CXCL10-expressing lymphocytes and cross-presenting dendritic cells. Altogether, this study provides a detailed phenotypic characterization of the immune cell subsets involved in cross-protection against ASFV, and highlights key functional immune mechanisms to be considered for the development of an effective ASF vaccine.

Omnibus (GEO; http://www.ncbi.nlm.nih.gov/geo/) under accession numbers GSE196472 and GSE196473. All data analysis scripts are available at https://github.com/funcgen/scRNAseq_ASFV.

**Funding:** This work has been funded by the Spanish Ministry of Science and Innovation [grant PID2019-107616RB-I00 (FR)], ISCIII/MINECO [grant PT17/0009/0019, cofunded by FEDER (AEC)], Huvepharma (FR, BG), and the Swedish Research Council for Environment [Agricultural Sciences and Spatial Planning (FORMAS), grant 2017-00486 (LL)]. Huvepharma participated in the study design of the first animal experiment.

**Competing interests:** The authors declare no competing interests.

## Author summary

African swine fever (ASF) pandemic is currently the number one threat for the porcine industry worldwide. Lack of treatments hampers its control, and the insufficient knowledge regarding the immune effector mechanisms required for protection hinders rational vaccine design. Here we present the first comprehensive study characterizing the complex cellular immune response involved in cross-protection against ASF. We show that, upon *in vitro* reactivation, cells from immune pigs induce a Th1-biased recall response that in turn enhances the antiviral innate response. Our results suggest that this positive feedback regulation of innate immunity plays a key role in the early control of ASF virus infection. Altogether, this work represents a step forward in the understanding of ASF immunology and provide critical immune components that should be considered to more rationally design future ASF vaccines.

## Introduction

African swine fever (ASF) is a contagious viral disease of domestic and wild pigs of mandatory declaration to the World Organisation for Animal Health (WOAH) (www.woah.org). The disease is caused by the African swine fever virus (ASFV), a large nucleocytoplasmic double-stranded DNA virus encoding more than 150 proteins [1]. In its most common clinical outcome, ASF pathogenesis is characterized by an acute hemorrhagic disease with high lethality reaching up to 100% [2,3]. The severity of the disease is to a large extent a consequence of the virus-induced cytopathic effect on monocytes and macrophages, and a marked lymphopenia affecting T and B cells [4]. In addition, the virus encodes several genes that modulate signaling pathways that regulate type I interferon (IFN-I) response, inflammation and apoptosis, altogether resulting in a rapid disruption of antiviral immune responses [5]. ASF has contributed to underdevelopment and poverty in affected areas of Africa during the last century, and the complex epidemiological situation in these regions has been a constant threat to unaffected countries [6]. Indeed, since its introduction from East Africa into Europe in 2007, the disease rapidly spread to many countries in Europe, Asia and Oceania, and more recently even reached the Caribbean, provoking massive economic losses to the swine industry [7]. Lack of an effective vaccine results in enormous difficulties to control virus spread, which mainly relies on rapid diagnosis and culling in affected farms (www.woah.org).

Experimental vaccines based on inactivated ASFV or subunit vaccine formulations have failed to induce solid protection [8–10], revealing the urgent need to increase our knowledge on ASF protective immunity [11]. In contrast, live attenuated viruses (LAVs) have emerged as potential ASF vaccines that could be used for emergency situations in affected countries. Indeed, several groups have obtained LAVs by the deletion of genes associated with virulence, which induce solid protective immunity against homologous strains [12–15]. Furthermore, our group developed a recombinant LAV lacking the CD2v protein (encoded by the EP402R gene), namely BA71ΔCD2. *In vitro*, BA71ΔCD2 infects monocytes/macrophages with similar efficiency than the parental virulent virus BA71 [16]. However, the virus is attenuated *in vivo* probably to its incapability to bind to red blood cells, a CD2v-mediated mechanism that ASFV uses to rapidly spread through the body [17]. Additionally, the cytoplasmic tail of CD2v has immunomodulatory functions [18], fact that might also contribute to the attenuation of the CD2v-depleted virus. BA71ΔCD2 is so far the only one described to confer cross-protection against the two virulent ASFV genotypes currently circulating in Europe and Asia (genotypes I and II) [16,19]. Although there is some concern due to LAVs-related biosafety problems

[20,21], the promising results obtained to date have resulted in an increasing interest for the development and optimization of new recombinant viruses [11,22]. Besides their potential as vaccine candidates, LAVs are important tools to study the protective ASFV-specific immune responses and to identify the viral antigens involved [23–25], two critical gaps for the rational development of ASF vaccines.

There is consensus that both humoral and cellular immune responses play a coordinated role in controlling virus expansion, but the precise underlying mechanisms remain elusive [26,27]. The beneficial role of antibodies was demonstrated by means of *in vivo* inoculation of immunoglobulins or colostrum from immune pigs, which resulted in partial protection of the recipient animals against a lethal challenge [28,29]. However, the existence of neutralizing antibodies is controversial [8,30], and only the presence of hemagglutination inhibitory antibodies capable of inhibiting infection *in vitro* has been correlated with protection [31,32]. Additionally, ASFV-specific antibodies have been associated with antibody-dependent infection enhancement [33]. T cell responses also play a relevant role in ASF immunity [26,34,35]. Importantly, *in vivo* depletion of CD8α+ cells abrogates protection in immune pigs [36], and expansion of virus-specific memory CD4 and cytotoxic CD8 T cells has been observed after *in vitro* stimulation of blood cells from immune pigs [37–39]. Nevertheless, only circulating ASFV-specific IFNγ-producing T cells have been associated with protection [25,34,40,41], and thus a phenotypic and functional characterization of ASFV-specific T cells induced after immunization is still lacking.

To gain better insight into the immune components involved in protection against ASFV, here we applied bulk and single-cell RNA-sequencing to characterize systemic and local *in vitro* recall responses induced in circulating and lymph node cells from BA71ΔCD2-immunized pigs. We show that intranasal vaccination with the BA71ΔCD2 LAV confers dose-dependent cross-protection against lethal challenge by direct contact with pigs infected with the pandemic Georgia2007/1 ASFV strain. This protection was associated with a broad recall immune response involving activation of both lymphocytes and myeloid cells. More specifically, *in vitro* ASFV-specific stimulation of cells from immune pigs revealed a robust Th1 response defined by the presence of polyfunctional CD4+CD8+ T cells as well as the expansion of cytotoxic T cells. Interestingly, this adaptive immune response induced an IFNγ-dependent positive feedback regulation of innate immunity, characterized by rapid activation of CXCL10-mediated inflammation, which in turn further enhanced Th1 responses. Altogether, these results unravel key immune components involved in a protective recall response against ASFV, suggesting a role of a timely adaptive-innate immunity crosstalk in the induction a rapid inflammatory response to efficiently control ASFV infection.

## Results

### Intranasal vaccination with BA71ΔCD2 confers dose-dependent cross-protection against a direct-contact challenge with infected pigs

We have previously demonstrated that intramuscular vaccination of pigs with the live attenuated BA71ΔCD2 ASFV confers protection against intramuscular challenge with homologous and heterologous virulent ASFV strains [16]. Aiming to optimize the vaccine efficacy against natural oronasal ASFV infection, here we tested its immunogenicity and cross-protective capacity after intranasal vaccination. Three groups of six pigs each were inoculated with three different BA71ΔCD2 doses, and one extra group of six pigs remained unvaccinated. Independently of the vaccine dose used, pigs did not show ASF-compatible clinical signs (S1 Fig). Presence of viral DNA was not detected neither in sera nor whole blood from animals receiving the high dose of BA71ΔCD2, except for one animal at day 21 p.v. showing virus loads by qPCR

under the threshold of quantification (S1 Table). Three weeks postvaccination (p.v.), animals from the four groups were challenged by direct contact with pigs infected with the virulent Georgia2007/1 strain (Fig 1A). All unvaccinated animals died during the second week after ASFV exposure showing clinical signs characteristic of ASFV acute infection such as prolonged fever and high virus loads in sera and nasal swabs (Figs 1B, S2 and S3), demonstrating the effectiveness of the lethal infection model used. In contrast, vaccinated pigs were protected in a dose-dependent manner. The high dose tested [$10^6$ plaque forming units (pfu) per pig] protected all animals from lethal challenge without showing major clinical signs (S2 Fig), while pigs inoculated with the intermediate ($3.3 \times 10^4$ pfu) and the low ($10^3$ pfu) vaccine doses were partially protected (Figs 1B and S2). Importantly, surviving animals vaccinated with the high and intermediate doses avoided virus expansion as demonstrated by the low Georgia2007/1 virus titers detected in sera and nasal cavities (S3A and S3B Fig, and S2 Table). The vaccine-induced systemic ASFV-specific humoral and cellular responses were also dose-dependent. All pigs vaccinated with the high and intermediate doses showed high levels of virus-specific antibodies from day 14 p.v., except for the only pig that did not survive (S3C Fig). In contrast, animals receiving the low vaccine dose showed significantly lower antibody levels. Similarly, only the high vaccine dose induced in all pigs IFNγ-secreting peripheral blood mononuclear cells (PBMC) at day 21 p.v., responding to *in vitro* stimulation with both the attenuated BA71ΔCD2 and the virulent Georgia2007/1 strains (Fig 1C). Importantly, overall vaccine-specific immune responses correlated with protection. Regardless of the vaccine dose used, most protected pigs showed significant levels of ASFV-specific antibodies and IFNγ-secreting cells, and both immune parameters inversely correlated with the appearance of high fever after challenge (Fig 1D).

## Vaccine-specific transcriptomic recall response to ASFV unravels concomitant Th1 and inflammatory signatures

To further investigate the systemic immunological memory induced by BA71ΔCD2 intranasal vaccination, we next aimed to characterize the recall response to ASFV upon *in vitro* stimulation. To this end, all the following experiments were performed using PBMC obtained from pigs three weeks after receiving the higher vaccine dose. Cells from unvaccinated and vaccinated pigs were stimulated with BA71ΔCD2 or left untreated for 10 hours, and levels of a panel of nine cytokines in supernatants were quantified by multiplex Luminex assay. IFNγ was not detected in supernatants, in line with the low number of ASFV-specific memory T cells in PBMC from BA71ΔCD2-vaccinated pigs as revealed by ELISpot (Fig 1B). In contrast, TNF and IFNα were the only two cytokines significantly produced by stimulated cells from vaccinated pigs, indicating a prominent role of innate immunity in the recall response to ASFV (Fig 1E).

We next characterized the PBMC transcriptomic signature associated with the *in vitro* recall response of four samples per group. Multidimensional scaling (MDS) analysis of the datasets obtained by RNA-sequencing (RNA-seq) revealed major transcriptional changes in BA71ΔCD2-stimulated samples from vaccinated pigs (S4A Fig). Indeed, taking unstimulated cells as a reference, we identified 2,176 differentially expressed (DE) genes in cells from vaccinated pigs, while only 144 DE genes were found in cells from unvaccinated animals, most of them shared between the two groups (Figs 2A and S4B). Gene Ontology (GO) analysis revealed that cells from both animal groups were enriched in genes involved in viral replication and IFN-I response (S4C Fig and S3 and S4 Tables), thus reflecting the innate immune response triggered by *in vitro* stimulation with the attenuated BA71ΔCD2 virus. However, in concordance with IFNα levels detected in supernatants (Fig 1E), expression levels of

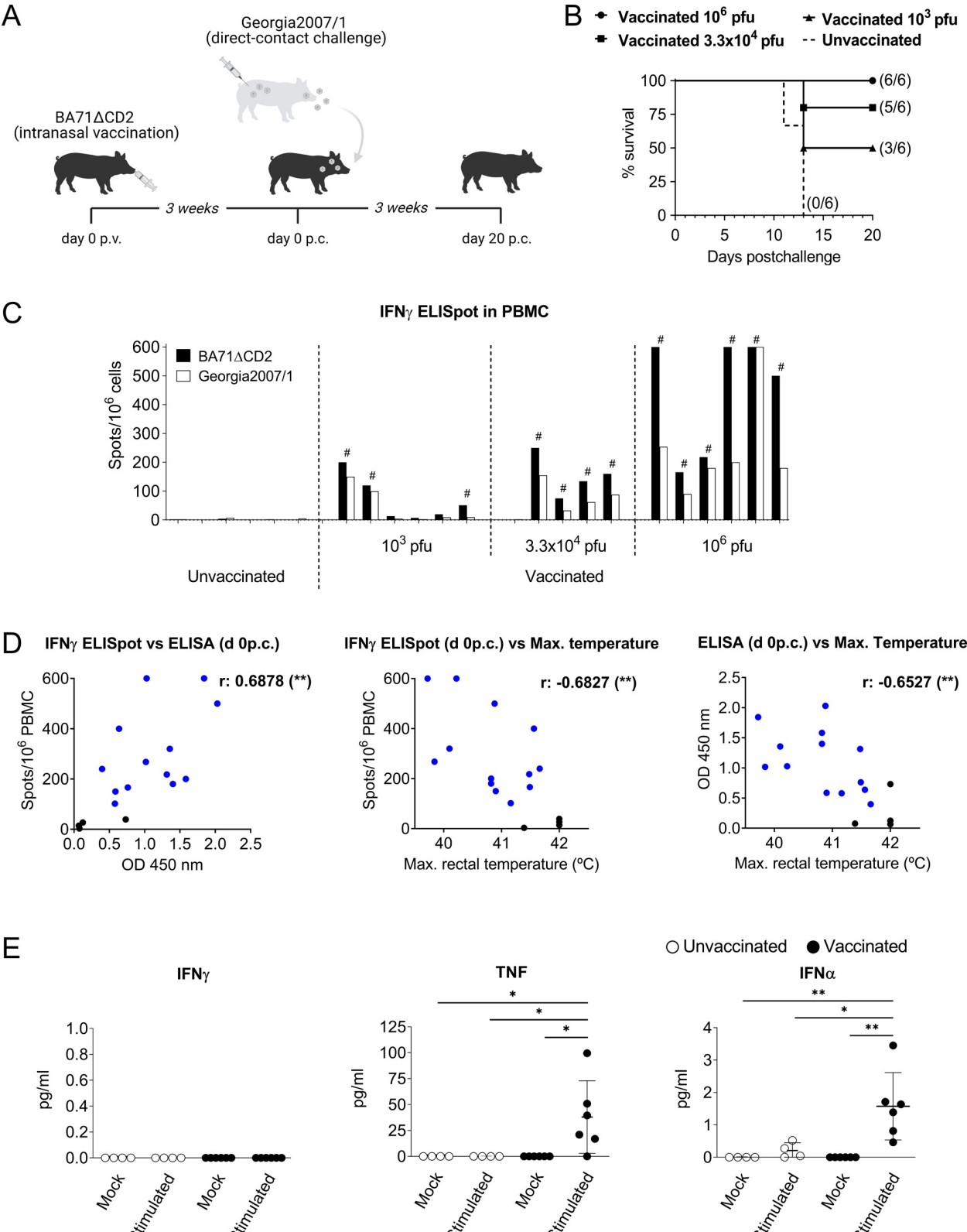

**Fig 1. Intranasal vaccination with BA71ΔCD2 induces dose-dependent protection associated with peripheral ASFV-specific immune responses.** (A) Schematic representation of the experimental design. (B) Survival plot showing the percentage of alive pigs at the indicated time points after contact with pigs infected with Georgia2007/1. The number of surviving pigs at the end of the experiment is indicated in brackets. (C) PBMC from day 21 p.v. were stimulated *in vitro* with BA71ΔCD2 or Georgia2007/1 and the number of ASFV-specific IFNγ-secreting cells was

quantified by ELISpot. Values shown are average values of duplicates subtracting the corresponding values of mock-stimulated cells. Octothorpes indicate surviving pigs. One animal vaccinated with the intermediate dose could not be evaluated due to a problem with the sampling. (D) Pearson's correlation between the number of ASFV-specific IFNγ-secreting cells (spots/$10^6$ PBMC) or antibody levels (OD 450 nm) and the maximum rectal temperature reached by each pig during the experimental infection. Blue and black dots represent pigs protected or unprotected against the lethal challenge, respectively. (E) Cytokine levels in culture supernatants of mock- or BA71ΔCD2-stimulated PBMC were quantified by Luminex-based multiplex assay. Vaccinated animals (n = 6) with the higher dose ($10^6$ pfu of BA71ΔCD2) are represented in bold symbols and control animals (n = 4) are represented in empty symbols. Statistical significance was determined by two-way ANOVA followed by Tukey's multiple comparisons test, and is displayed in GraphPad style (p > 0.05 ns, * p ≤ 0.05, ** p ≤ 0.01).

interferon-stimulated genes (ISG) were significantly higher in PBMC from vaccinated pigs than from unvaccinated ones (Fig 2B), suggesting a vaccine-mediated enhancement of innate immunity during the recall response. GO analysis of the genes specifically deregulated only in cells from vaccinated pigs revealed the induction of T and B cell responses (S4C Fig), indicating the presence of circulating ASFV-specific memory lymphocytes. A detailed analysis of the corresponding genes suggested the activation of a Th1-biased recall response in these samples, as shown by the upregulation of the two key cytokines *IFNG* and *TNF* (Fig 2C), as well as others such as *IL2*, *IL15*, *IL27* and *XCL1*, and the T cell activation markers *CD69* and *CD274* (Fig 2B). Interestingly, together with this adaptive immune response, we also found a robust innate immune signature represented by terms such as NF-kB- and MAPK-signaling, TNF production, and macrophage differentiation (S4D Fig and S4 Table). To note, expression patterns of the genes within these terms, such as the upregulation of *BATF2*, *IRF1*, *CXCL9*, *CXCL10*, *CCL2* and *CCL4*, as well as the downregulation of *CD163* and *CCR2* (Figs 2B and S4D), suggested monocytes/macrophages as the main cell subsets involved in this inflammatory response [42–44].

The transcriptional signature identified so far was in response to *in vitro* stimulation with the vaccine virus BA71ΔCD2, and thus might not be indicative of the immune responses induced in vaccinated pigs after infection with the heterologous virulent strain. Thus, we next aimed to validate the results obtained so far stimulating cells with the Georgia2007/1 ASFV used for the lethal *in vivo* challenge. A group of 8 pigs were intranasally vaccinated with $10^6$ pfu of BA71ΔCD2 following the same immunization regimen described above (Fig 1A), and 6 pigs were used as unvaccinated controls. Three weeks later, fresh PBMC were isolated and *in vitro* stimulated either with BA71ΔCD2 or Georgia2007/1 ASFV. To assess the transcriptional responses induced at 10 hours post-stimulation, we selected 37 DE genes from the RNA-seq dataset representative of both the inflammatory and the Th1 immune responses previously observed, and quantified their expression levels by a microfluidic quantitative PCR assay. Importantly, stimulation with both virus strains resulted in very similar gene expression patterns, characterized by the upregulation of ISG in samples from both animal groups, and the specific deregulation of Th1- and inflammatory-related genes in cells from vaccinated pigs (Fig 3 and S5 Table). Altogether, these results demonstrate that intranasal vaccination of pigs with BA71ΔCD2 induce a systemic cross-reactive Th1 memory response that is linked with a rapid enhancement of innate immunity upon *in vitro* activation.

## IFNγ from polyfunctional ASFV-specific T cells triggers TNF production by myeloid cells

Two hypotheses might explain the robust inflammatory signature observed during the recall response to ASFV in PBMC from vaccinated animals: (i) a higher virus replication during the *in vitro* stimulation, and/or (ii) an IFNγ-mediated activation of monocytes. The first hypothesis might be explained by the presence of higher percentages in blood from vaccinated pigs of differentiated macrophages, which are more susceptible to ASFV infection [45–47]. However, identical

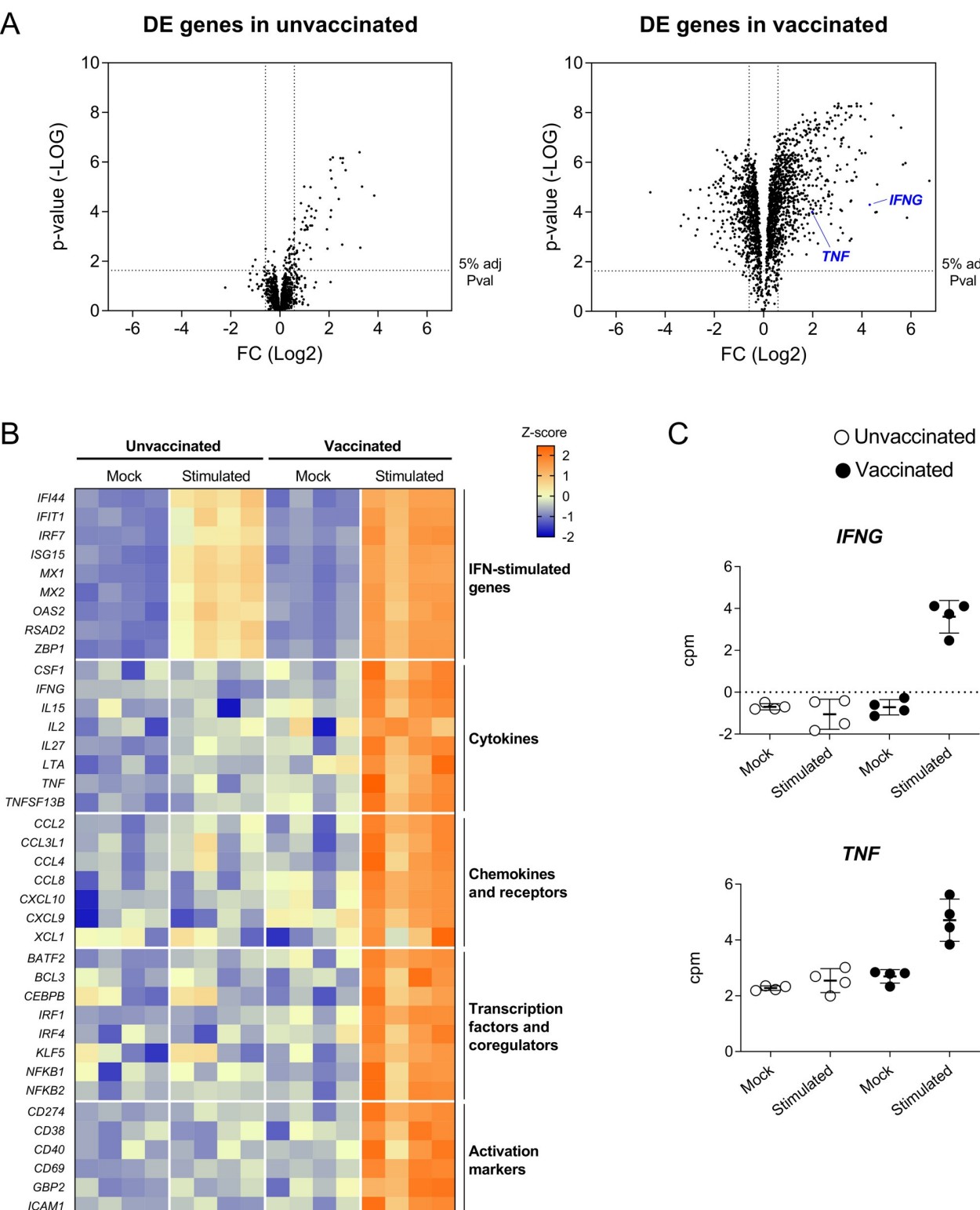

**Fig 2. DE genes in PBMC from BA71ΔCD2-vaccinated pigs after *in vitro* ASFV-specific stimulation reveal concomitant Th1 and inflammatory responses.** (A) Volcano plots showing fold changes and adjusted p-values for genes differentially expressed between unstimulated (mock) and ASFV-stimulated cells from unvaccinated and vaccinated pigs. (B) Heatmap depicting normalized RNA-seq-derived log2-counts per million (log2CPM) values of representative DE genes. Normalized values from unstimulated (mock) and ASFV-stimulated cells from unvaccinated and vaccinated pigs are shown. (C) RNA-seq-derived expression levels of *IFNG* and *TNF* as log2CPM values.

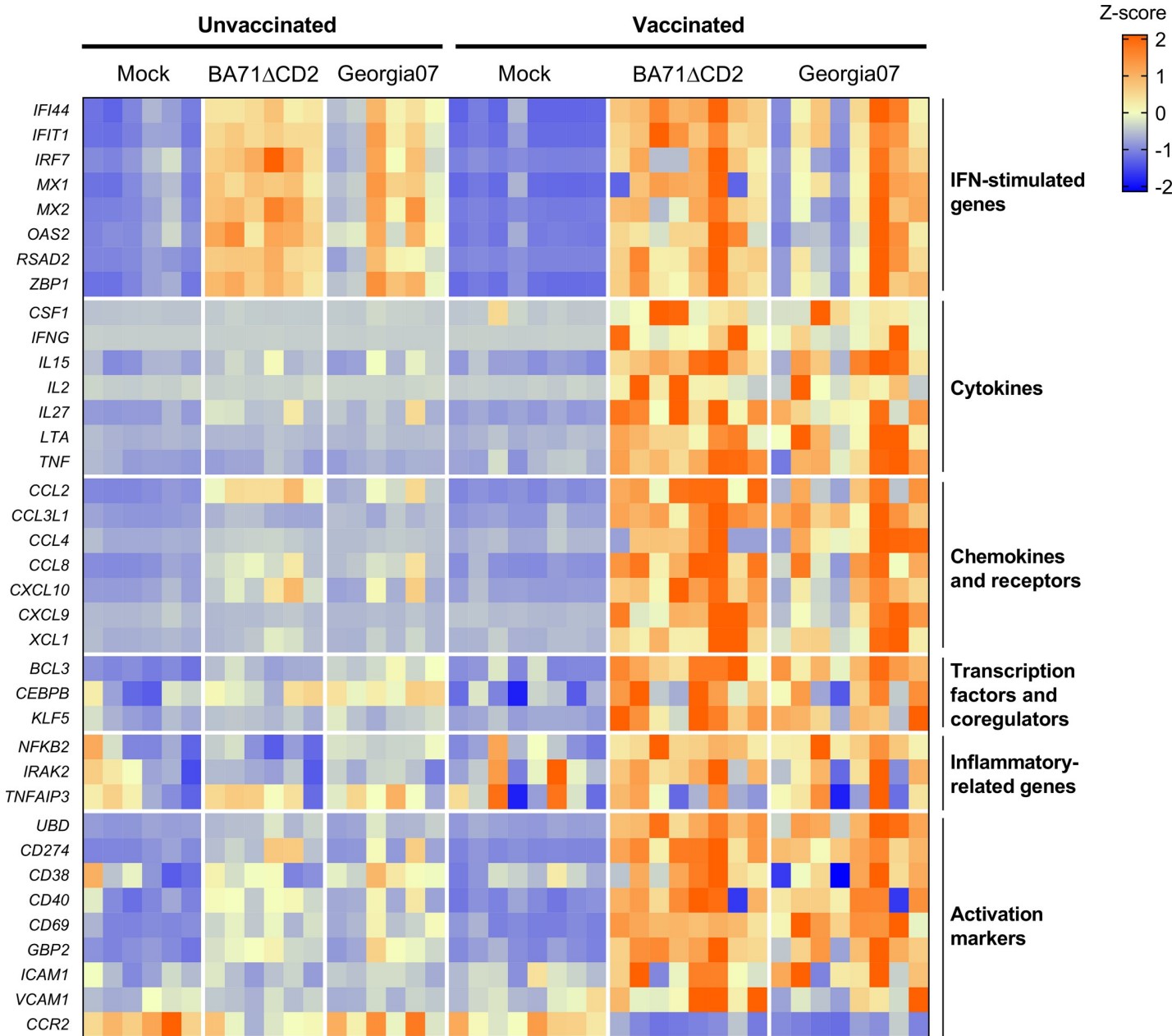

**Fig 3. BA71ΔCD2 vaccination induces a similar recall transcriptomic response upon *in vitro* stimulation with BA71ΔCD2 and the heterologous Georgia2007/1 ASFV.** PBMC from BA71ΔCD2-vaccinated or unvaccinated pigs were stimulated with BA71ΔCD2, Georgia2007/1 or left untreated (mock), and expression levels of 37 genes representative of the results obtained from the RNA-seq dataset were quantified by microfluidic quantitative PCR assay. The heatmap shown illustrates normalized gene expression levels.

percentages of total (CD3-CD172a+) and mature myeloid cells (CD3-CD172a+SLAII+ or CD3-CD163+) were found in PBMC from unvaccinated and BA71ΔCD2-vaccinated animals as assessed by flow cytometry (Fig 4A). In addition, analysis of RNA-seq-derived viral transcripts in PBMC after *in vitro* stimulation showed indistinguishable expression levels of both early and late viral genes in ASFV-stimulated PBMC from both animal groups (odds ratio: 0.9569) (Fig 4B). Thus, these results suggested that the inflammatory recall response was triggered by IFNγ from

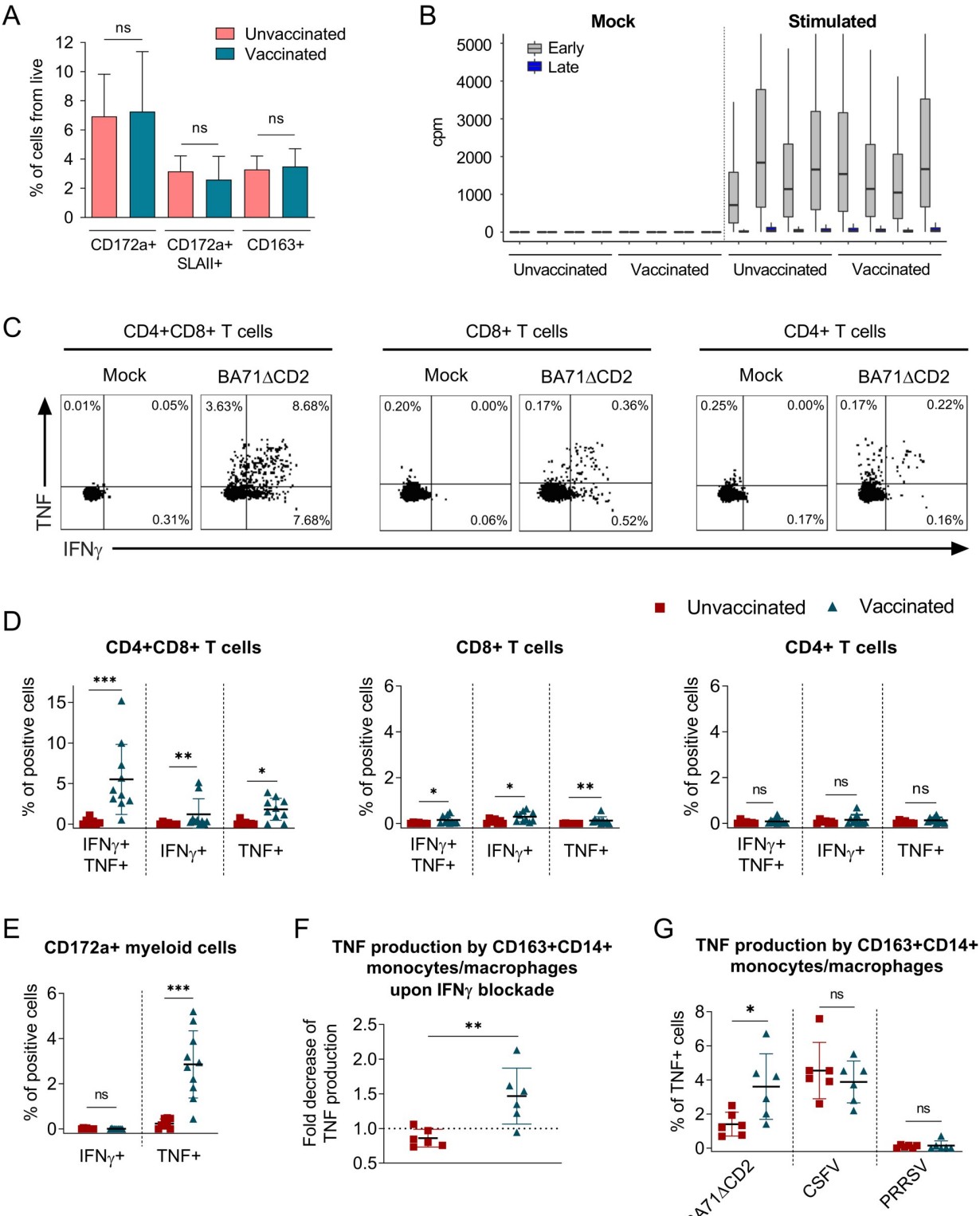

**Fig 4. IFNγ from ASFV-specific CD4+CD8+ T cells induces TNF-production by myeloid cells.** (A) Percentage of myeloid cells (CD3-CD172a +) and monocytes/macrophages (CD3-CD172a+SLAII+ or CD3-CD163+) in PBMC from unvaccinated (n = 5) and BA71ΔCD2-vaccinated (n = 7) pigs assessed by flow cytometry. (B) RNA-seq-derived number of reads mapping to early and late ASFV genes in PBMC after 10 hours stimulation with mock or BA71ΔCD2 ASFV. (C) Representative dot plots for intracellular IFNγ and TNF staining on T cells from a vaccinated pig. (D-E) Percentages of IFNγ- and/or TNF-producing CD4+CD8+, CD8+ and CD4+ T cells (D) or myeloid cells (CD3-CD172a+) (E) in

PBMC from unvaccinated (n = 7–8) and vaccinated (n = 10) pigs after mock or BA71ΔCD2 stimulation. (F) Fold decrease of TNF production by monocytes/macrophages (CD3-CD14+CD163+) in BA71ΔCD2-stimulated PBMC from unvaccinated (n = 6) and BA71ΔCD2-vaccinated (n = 6) animals treated or untreated with anti-IFNγ antibody. The fold decrease was calculated by dividing percentages of TNF-producing cells without IFNγ blockage by percentages of TNF-producing cells with IFNγ blockage. (G) Percentages of TNF-producing monocytes/macrophages (CD3-CD14+CD163+) in PBMC from unvaccinated (n = 6) and BA71ΔCD2-vaccinated (n = 6) animals upon stimulation with BA71ΔCD2, CSFV or PRRSV. Statistical significance was determined by unpaired two-tailed t-test for normally distributed data, or two-tailed Mann-Whitney U test for not normally distributed data, and is displayed in GraphPad style (p > 0.05 ns, * p ≤ 0.05, ** p ≤ 0.01, *** p ≤ 0.001).

virus-specific memory T cells. To test this hypothesis, we first immunophenotyped the cell subsets involved in IFNγ- and TNF-secretion (Fig 4C), two key representative cytokines of the BA71ΔCD2-induced immunity (Fig 2). Intracellular cytokine staining of stimulated PBMC showed elevated percentages of vaccine-specific IFNγ- and TNF-producing CD4+CD8+ T cells (Fig 4C and 4D), a phenotype characteristic of porcine memory T cells [48]. However, in concordance with the results obtained by IFNγ ELISpot (Fig 1C), percentages of ASFV-specific CD4+CD8+ T cells were low and therefore unlikely to be the only source of TNF observed in cell culture supernatants (Fig 1E). Indeed, myeloid cells from vaccinated pigs produced TNF, but not IFNγ, in response to BA71ΔCD2 stimulation (Fig 4E), thus demonstrating their contribution to the vaccine-specific inflammatory response. Importantly, this TNF production was significantly reduced in CD14+CD163+ monocytes/macrophages by blocking of IFNγ with a specific antibody (Fig 4F). We next evaluated whether the inflammatory response observed only in cells from vaccinated pigs may be due to the induction of trained immunity in blood monocytes after vaccination with the BA71ΔCD2 LAV [49]. With this purpose, we stimulated PBMC from vaccinated and unvaccinated pigs with the attenuated ALL-183 strain of porcine reproductive and respiratory syndrome virus (PRRSV) or the virulent Margarita strain of classical swine fever virus (CSFV), two viruses also infecting monocytes and/or macrophages. While stimulation with CSFV induced TNF production by CD14+CD163+ monocytes/macrophages from both vaccinated and unvaccinated pigs, stimulation with PRRSV did not trigger this inflammatory response (Fig 4G). Furthermore, we confirmed the specificity of the inflammatory response in PBMC from vaccinated animals upon stimulation with BA71ΔCD2 (Fig 4G). Overall, these results suggest that the inflammatory recall response induced in PBMC from vaccinated pigs is dependent on IFNγ production from ASFV-specific polyfunctional memory T cells.

## scRNA-seq analysis of lymph node cells unmasks a cytotoxic recall response and confirms the Th1-dependent inflammatory signature

The induction of immunological memory at the sites where viruses initially replicate during a natural infection is critical for protection. Thus, we next investigated whether the *in vitro* recall responses to ASFV observed in PBMC from intranasally vaccinated pigs were also induced in submandibular lymph node (LN) cells, one of the first target tissues after an oronasal ASFV infection [50]. With this aim, we performed a new vaccination experiment where pigs were inoculated with $10^6$ pfu of BA71ΔCD2 and sacrificed three weeks postvaccination (S5 Fig). Indeed, most vaccinated pigs showed high levels of ASFV-specific IFNγ-producing cells in submandibular lymph node as measured by ELISpot (S6A Fig), while low or undetectable levels of viral DNA were found (S6 Table). To further characterize this vaccine-specific response, we performed single-cell RNA-sequencing (scRNA-seq) of LN cells from a control and a vaccinated animal. Cell suspensions obtained from the digested tissues were *in vitro* stimulated for 16 hours with BA71ΔCD2, and methanol-fixed prior sequencing using the 10x Genomics scRNA-seq platform. A total of 4,254 and 5,849 cells were analyzed in samples from the unvaccinated and the vaccinated pig, respectively. We identified 22 transcriptionally distinctive clusters which were assigned to different cell subsets based on canonical lineage markers (Fig 5).

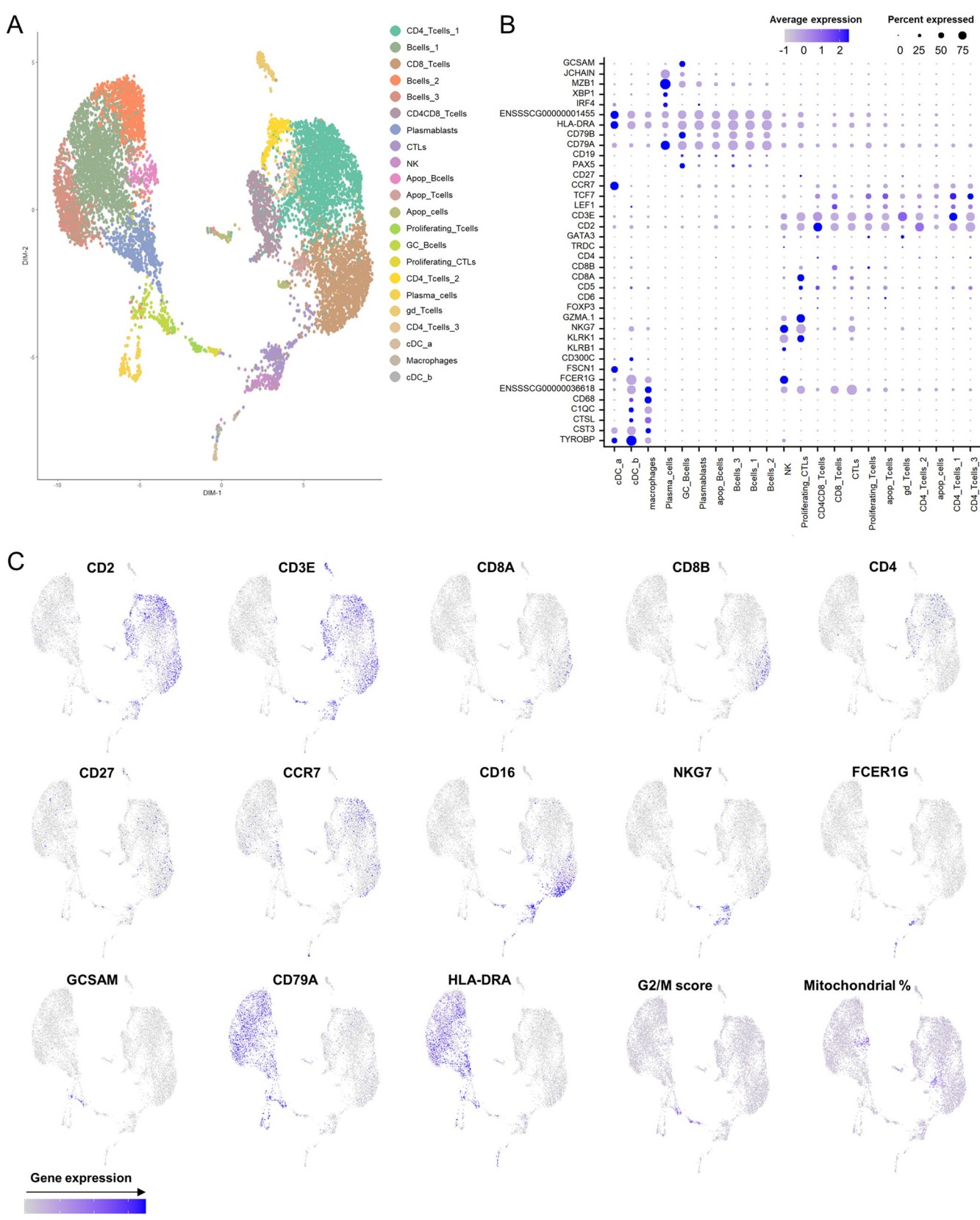

**Fig 5. Classification of scRNA-seq clusters from porcine submandibular LN cells.** (A) Uniform Manifold Approximation and Projection (UMAP) plot representing 22 transcriptionally distinctive clusters, each one differently colored. (B) Dot plot visualization of selected marker genes in each cluster. Dot size represents the percentage of cells expressing the particular gene, while the spectrum of color indicates the mean expression levels. *ENSSSCG00000001455 = SLA-DRB1, ENSSSCG00000036618 = FCGR3A.* (C) Feature plots depicting single-cell expression of key cluster-defining genes.

These included macrophages, dendritic cells (DCs), natural killer (NK) cells (*NKG7+KLRB1+ KLRK1+*), CD4 and CD8 T cells (*CD3E+CD4+* and *CD3E+CD8A+CD8B+*, respectively), cytotoxic CD8 T cells (CTLs) (*CD8A+GZMA.1+GZMK+*), proliferating CTLs and T cells (showing high expression of G2/M phase markers), γδ T cells (*TRDC+*), B cells (*CD79A+*), plasmablasts (*NME2+*), GC B cells (*GCSAM+*) and plasma cells (*JCHAIN+MZB1+XBP1+*) (Fig 5). Three B cell clusters were transcriptionally distinguished, but the corresponding cell subtypes could not be annotated. Clusters exhibiting a high percentage of mitochondrial content were designated as apoptotic cells (Fig 5).

We next analyzed the DE genes between samples to identify vaccine-specific cellular transcriptomic signatures. Only cell clusters with a high number of cells showed a significant number of DE genes (S6B–S6D Fig), and thus we restricted further analysis to them. Genes upregulated in all cell subsets were enriched in terms related to IFN-I and IFNγ responses (S7 Fig), thus validating the results obtained in PBMC. Indeed, ISG were significantly upregulated in several cell subsets from the vaccinated pig when compared to the control one (S8 Fig). Importantly, ASFV transcripts were predominantly found in macrophages and did not differ between the two LN samples, confirming that the enhanced innate immunity observed cannot be attributed to differences in the replication rate of the virus used for stimulation.

To further identify vaccine-specific features that might be associated with protection we next compared the relative number of cells in each cluster between the two samples (Fig 6). Several populations were significantly overrepresented in LN cells from the vaccinated pig (S7 Table), suggesting their contribution to the ASFV-specific recall response. These included several B cell subsets such as plasmablasts and plasma cells, an undefined CD4 T cell subset (CD4_Tcells_2), and γδ T cells (Fig 6). Interestingly, the proinflammatory CXCL10 chemokine was strongly upregulated in plasmablasts, in the undefined CD4 T cell subset (CD4_Tcells_2), and in cross-presenting DCs (cDC_a) (Fig 7A), thus further validating the induction of a Th1-biased recall response. Of particular interest was the presence of a robust vaccine-specific cytotoxic response characterized by elevated numbers of responding CD8 T cells (CTLs and proliferating CTLs) (Fig 6). In the sample from the vaccinated pig, CTLs showed downregulation of *CCR7* and *CXCR4*, a hallmark of differentiation to effector CD8 T cells [51], and profilin 1 (*PFN1*), a negative regulator of lytic granules release [52] (S9A Fig). Indeed, CTLs significantly upregulated *GZMA.1*, confirming their activated status, while it was similarly expressed in NK cells from both samples (Fig 7B). This ASFV-specific cytotoxic response was not detected during the *in vitro* recall transcriptomic response in PBMC (S9B Fig), which were stimulated shorter than LN cells. To demonstrate the induction of a systemic vaccine-specific cytotoxic response, we next analyzed perforin production in PBMC after a 48 hours stimulation, since monitoring of memory cytotoxic activity usually requires a long stimulation for reactivation and expansion in culture [53]. Importantly, cells from vaccinated animals showed a vaccine-specific expansion of perforin+ CD4+CD8+ T cells, and in a lesser extent also of perforin+ γδ T cells (Fig 7C). Interestingly, perforin+ NK cells and CD8+ T cells were increased in cells from both animal groups, indicating their contribution to the unspecific innate immune response induced during ASFV infection. However, their fold-increase was significantly higher in immune pigs (Fig 7C), suggesting that vaccine-induced responses enhance the cytotoxicity activity mediated by these two cell subsets. Altogether, evidence shown here

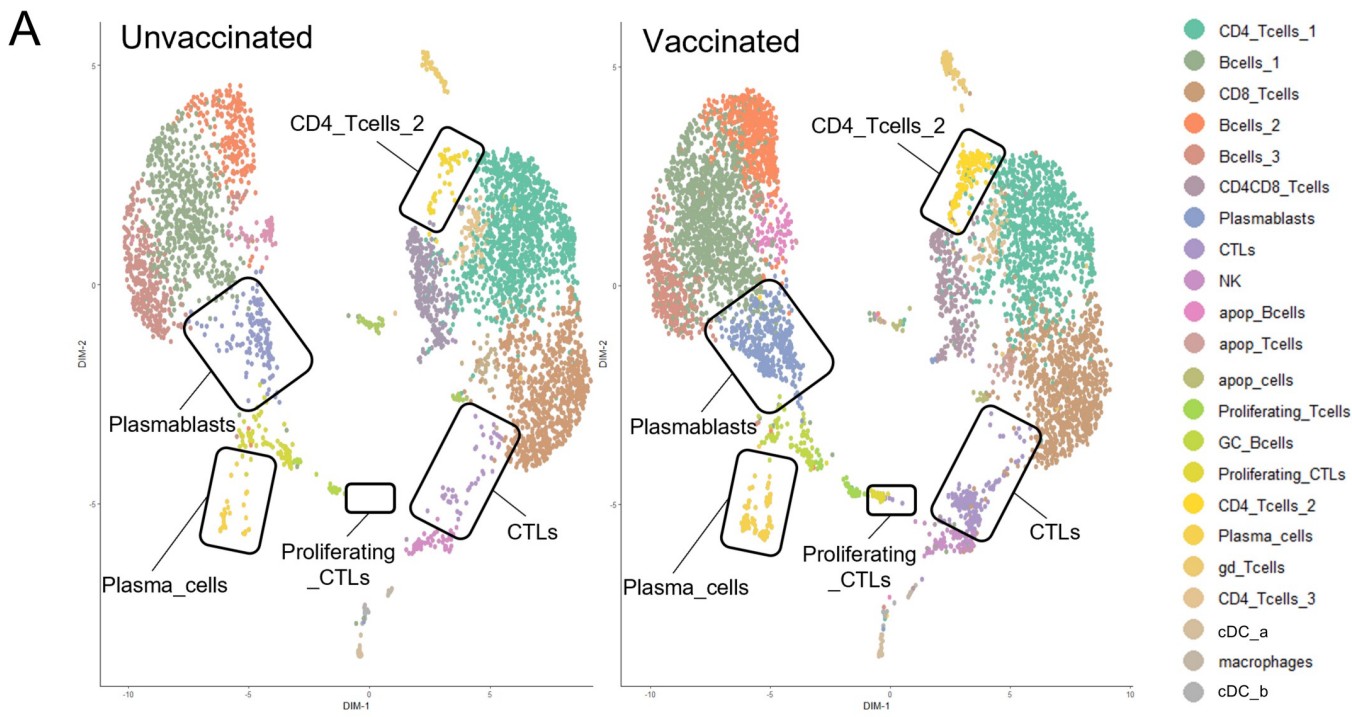

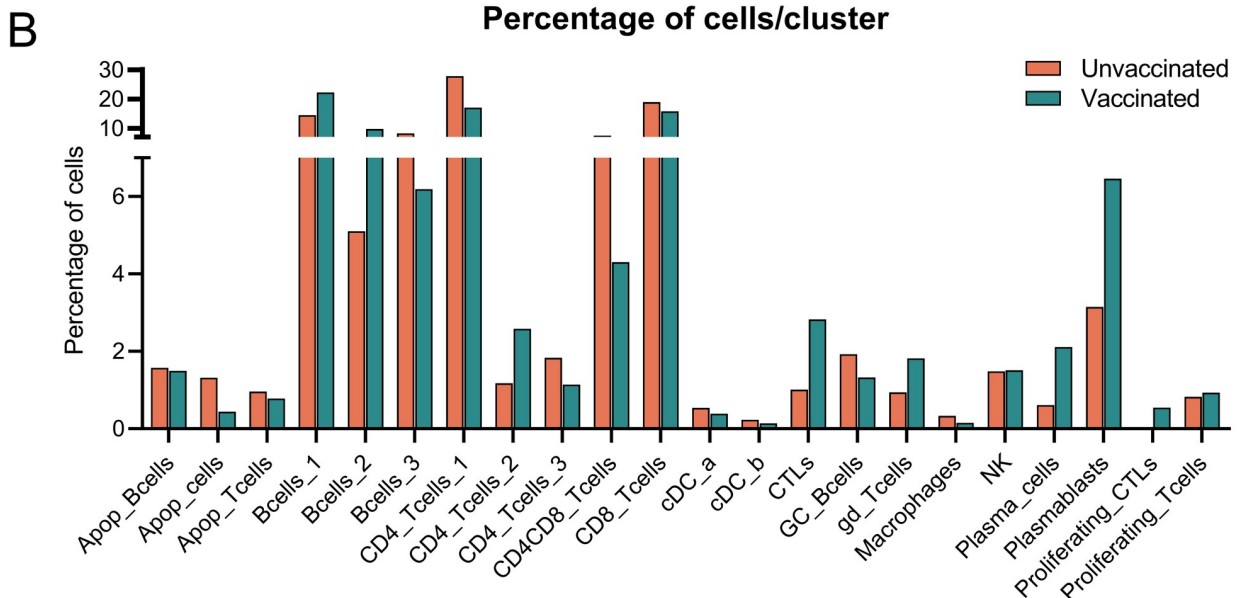

**Fig 6. Several cell subsets are involved in the ASFV recall response from submandibular LN cells.** (A) UMAP plots comparing cell clustering between unvaccinated and vaccinated samples. Cell clusters showing major differences among the two samples are highlighted. (B) Comparison of the percentage of cells in each cluster between samples.

demonstrates that the recall response to ASFV in vaccinated pigs promotes a crosstalk between adaptive and innate immunity characterized by a rapid triggering of concomitant inflammatory and cytotoxic responses.

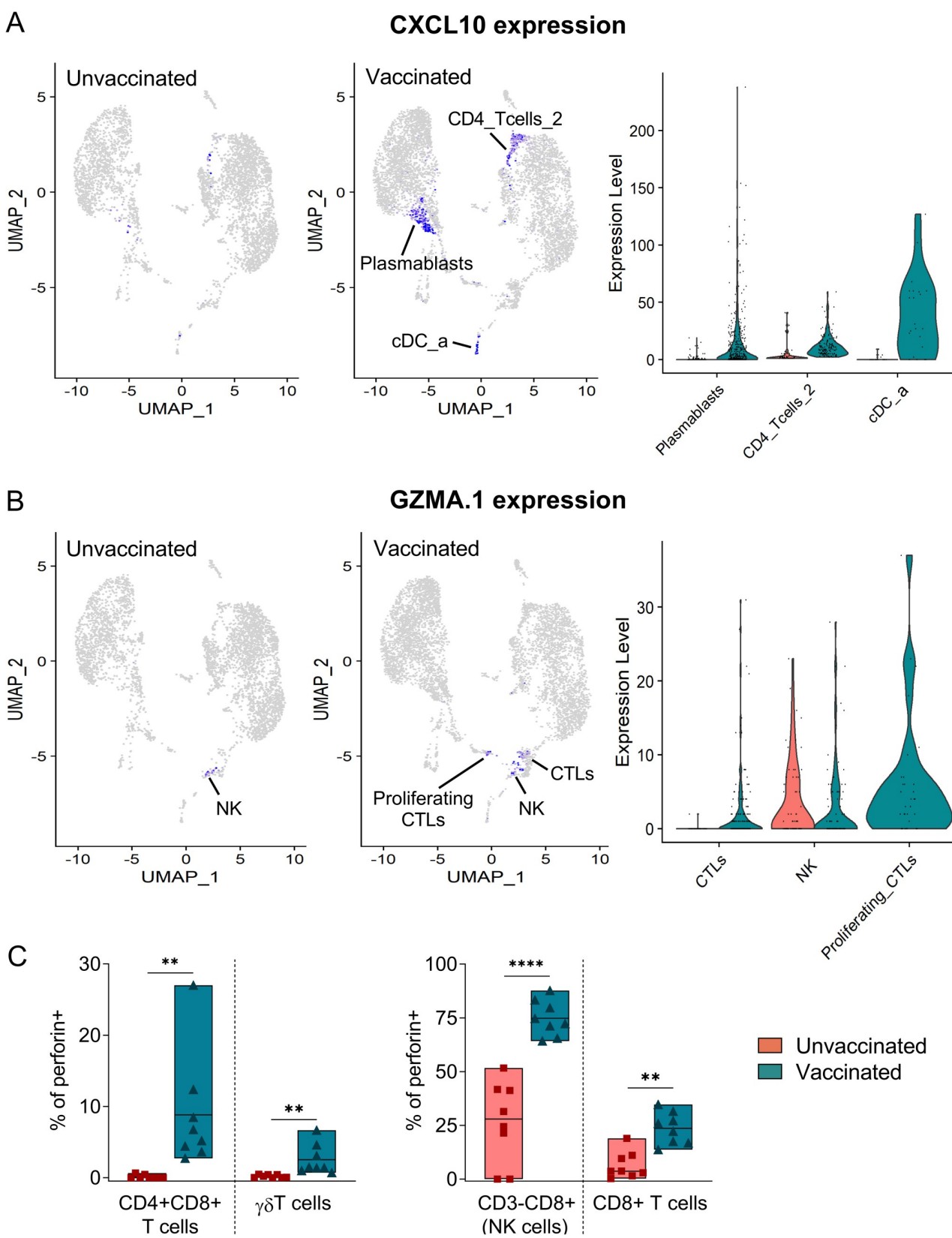

**Fig 7. scRNA-seq analysis reveals a CXCL10-driven robust cytotoxic recall response.** (A) UMAP and violin plot showing expression levels of CXCL10 in plasmablasts, cDC_a and CD4_Tcells_2. (B) UMAP and violin plot showing the expression levels of GZMA.1 in CTL and NK cells. (C) PBMC from unvaccinated (n = 8) and BA71ΔCD2-vaccinated (n = 8) pigs were stimulated with BA71ΔCD2 for 48 hours, and the percentages of CD8 +, CD4+CD8+, γδ T cells, and CD3-CD8+ cells (NK cells) producing perforin were assessed by flow cytometry. Statistical significance was determined by unpaired two-tailed t-test and is displayed in GraphPad style (p > 0.05 ns, ** p ≤ 0.01, **** p ≤ 0.0001).

## Discussion

The development of an effective vaccine against ASFV to control the current pandemic is partially hampered by the poor knowledge on ASF protective immunity [7,11]. To address this gap, here we provide a comprehensive study of the recall response to ASFV in circulating and lymph node cells from immune pigs. We demonstrate that intranasal vaccination with the attenuated virus BA71ΔCD2 confers a dose-dependent cross-protection against a lethal ASFV infection, which is associated with the presence of virus-specific polyfunctional and cytotoxic CD4+CD8+ T cells. *In vitro* reactivation of these cells revealed an IFNγ-dependent activation of an inflammatory response. This positive feedback regulation from adaptive to innate immunity resulted in the enhancement of virus-induced type I interferon response, a rapid differentiation of blood monocytes to activated macrophages, and an increase of nonspecific cytotoxic NK and CD8+ T cells. These data suggest that a prompt vaccine-induced activation of innate immunity as well as a broad cytotoxic response during the first hours of infection are critical immune components for protection against ASFV.

Our finding on the IFNγ-dependent activation of innate immunity during the recall response to ASFV is in accordance with previous studies in other infectious models [54–57]. These works demonstrated that antigen-dependent activation of memory T cells triggers an inflammatory response, which in turn further recruits innate immune cells and virus-specific memory T and B cells to the site of infection [54–57]. Similarly, our results show that a low number of polyfunctional ASFV-specific memory CD4+CD8+ T cells can trigger a robust innate immune response upon ASFV-specific stimulation. This innate response is characterized by the rapid upregulation of several cytokines and chemokines that orchestrate an inflammatory response. Importantly, the adaptive to innate immune crosstalk is reflected by the dramatic reduction of TNF-producing macrophages when blocking IFNγ, and suggested by the upregulation of the IFNγ-inducible chemokine CXCL10. Nevertheless, we cannot discard a potential role of vaccine-induced trained immunity in this enhanced response of myeloid cells [58]. Indeed, other studies have demonstrated that vaccination with certain live vaccines induce protection against non-related pathogens through the induction of innate immune memory [49]. However, blood monocytes/macrophages from vaccinated and unvaccinated pigs responded equally to infection with PRRSV and CSFV, thus indicating a lack of vaccine-induced innate immune memory in these cells. Further studies are required to investigate the potential induction of trained immunity in other tissues. For instance, it is well described the acquisition of trained immunity in alveolar macrophages [59], which might be especially relevant for an intranasal live vaccine targeting macrophages such as BA71ΔCD2. Altogether, our results suggest a key role of a prompt adaptive immunity-dependent enhancement of the antiviral innate response to control ASFV infection, as described for other models [55–57]. The relative contribution of this early innate immunity to the protection afforded against ASFV infection *in vivo* is an important issue that requires further investigation, and should also be evaluated in other relevant infections.

It is reasonable to speculate that an enhanced innate immunity during the ASFV recall response may benefit early control of virus expansion. For instance, although ASFV encodes several genes that interfere with IFN-I signaling [60,61], the rapid boost of IFN-I response and

the subsequent production of ISG will likely hinder virus spread to neighboring cells. In addition, IFNα- and IFNγ-activated macrophages are more resistant to ASFV infection [62–64], and thus their proximity to IFNγ produced by effector memory T lymphocytes would reduce the number of susceptible cells. Moreover, macrophages have a major role in ASF pathogenesis both by direct and bystander effects [5]. Therefore, their rapid activation during the recall response before becoming infected would allow their active participation in the antiviral effector immunity, which is compromised during ASFV infection in nonimmune pigs [65]. Altogether, this global antiviral state induced at the site of virus entry would facilitate resolution of the infection. However, our transcriptomic data from stimulated PBMC clearly indicated that blood monocytes differentiate to macrophages, which are more susceptible to ASFV infection [45]. Thus, we cannot discard that activation of innate immunity during the first hours of infection might favor local ASFV replication by increasing the number of target cells. Our *in vitro* data showed similar virus replication rates in samples from vaccinated and unvaccinated pigs after a short stimulation, and we did not observe any *in vivo* evidence of higher virus titers in vaccinated pigs at early time points postchallenge. Nonetheless, new experiments would be necessary to properly address this issue. Furthermore, the balance between the antiviral innate immune responses and the recruitment of target cells to the site of infection during recall responses may be critical for other viruses targeting immune cells, such as classical swine fever virus (CSFV) and porcine reproductive and respiratory syndrome virus (PRRSV) in pigs, and human immunodeficiency virus (HIV) in humans.

We also observed activation of several cytotoxic cell subsets during the *in vitro* recall response. First, percentages of perforin-producing NK and single CD8+ T cells were significantly higher in samples from vaccinated pigs. These nonspecific cells are likely responding to ASFV primary infection, since they are also expanded within cells from unvaccinated pigs after *in vitro* virus stimulation. Indeed, the presence of both cytotoxic cell subsets in blood has been previously associated with control of ASFV systemic infection [26,36,66]. Importantly, our data indicates that the expansion of these nonspecific cytotoxic cells is enhanced through the activation of the vaccine-induced IFNγ-CXCL10 communication axis at the site of virus entry, and might be critical to hinder early viral replication. Second, we demonstrate the specific stimulation of perforin-producing CD4+CD8+ memory T cells in vaccinated pigs, which matches the observed Th1 signature. Nevertheless, in contrast to NK and single CD8+ T cells, these memory T cells were only identified after long *in vitro* stimulation, further indicating their antigen-specificity. This result validates preliminary results from Takamatsu et al. associating the proliferation of double positive CD4+CD8+ CTLs after *in vitro* stimulation with ASF protection [26]. To note, the three cytotoxic cell subsets mentioned above (NK cells, single CD8+ T and double positive CD4+CD8+ T cells) express the CD8α protein. Thus, all of them might participate in the protection afforded as demonstrated by the critical role of CD8α + cells after their *in vivo* depletion in immune pigs [36], and further studies are required to evaluate their relative importance. Finally, concomitant with cytotoxic CD4+CD8+ T cells, we also identified the specific expansion of perforin-producing γδ T cells in PBMC from vaccinated pigs. Porcine γδ T cells are abundant in blood but do not express perforin in basal conditions [67], and thereby the increase of their cytotoxic activity upon ASFV stimulation might be relevant. Since γδ T cells recognize different types of antigens than αβ T cells, they might complement the cytotoxic response mediated by CD4+CD8+ memory T cells. Altogether, we demonstrate that vaccinated pigs respond to ASFV stimulation by the activation of a broad cytotoxic response, further confirming and expanding the importance of cellular immunity during ASFV infection [26,34].

The future identification of correlates of protection against ASFV infection will be crucial to understand ASF immunity and to evaluate vaccine efficacy. Even though in the present

study we observed an overall correlation of both antibody titers and IFNγ-producing cells with the protection afforded, these two parameters do not always correlate with protection [16]. Lack of classical neutralizing antibodies in immune pigs complicates the standardization of antibody-based assays which may help define thresholds of functional antibodies required for protection [68]. The functional characterization of ASFV-specific Th1 and cytotoxic cells induced after vaccination opens the opportunity to evaluate their contribution as correlates of protection. Indeed, we have previously associated proliferating cross-reactive CD8 T cells with heterologous protection in BA71ΔCD2 vaccinated pigs [16]. However, this would require an ambitious study analyzing the immune status of a high number of pigs. Moreover, it is plausible that protection against ASFV requires the coordination of several immune components, and that variable levels of each of them will determine the thresholds required to control infection, as shown in other infections [57,69]. Indeed, both antibodies and CD8 T cells are known to contribute to protection against ASFV [28,29,36], and thus a proper quantification of the corresponding functional mechanisms will be necessary to establish correlates of protection. Additionally, as demonstrated in other infection models [57,70], protection against ASFV might be associated with local immunity induced at the site of virus entry, which is not necessarily measurable in blood. The results obtained in this study illustrate the complexity of the cellular responses during ASFV recall response, and their basic immune components were identified both in PBMC and cells from submandibular lymph node, one of the first tissues affected by ASFV during natural infection [50]. Consequently, it will be important to investigate whether transcriptomic signatures obtained from *in vitro* stimulated PBMC might unravel expression thresholds of representative genes allowing to discriminate protected from unprotected pigs.

Despite the efficacy of ASF LAVs in experimental conditions, their use in the field entails biosafety risks [21,22]. Vaccine candidates are tested in healthy animals, and thus do not cover the risk of using LAV in immunocompromised pigs that might be present in the field. For instance, two recent studies have demonstrated that susceptibility to ASFV infection varies depending on pigs' immune status [35,71,72]. Therefore, ASF live attenuated vaccines should comply with minimal efficacy and safety requirements implemented by the competent regulatory agencies. Finally, it is important to mention the relevance of the model used in this study. BA71ΔCD2 (genotype I) is the only live attenuated vaccine prototype that confers cross-protection against experimental challenge with different ASFV genotypes, including the genotype II pandemic virus. Therefore, the comparison of our results with the immune responses induced with other live attenuated viruses will allow the identification of the immune parameters required for cross-protection. Interestingly, to induce cross-protection against viruses from the distant genotype IX, BA71ΔCD2 requires a boost with the homologous virulent BA71 strain [24], as occurs using other attenuated ASFV [24,40]. Although the immune mechanisms boosted during the secondary infection are unknown, innate immunity might play a crucial role to generate a sufficient and effective adaptive immune response, as demonstrated for booster vaccination against SARS-CoV-2 [73,74]. Thus, the cross-protecting innate immune signature that we have observed during *in vitro* recall response to ASFV might represent a general requirement to achieve protective ASF immunity. Lastly, the challenge by direct contact with infected pigs used in this experimental study mimics the most common infection route among wild boars and domestic pigs. Therefore, mucosal immunity induced by intranasal vaccination with BA71ΔCD2 proved to be effective for the control of ASFV spread in the field. Nonetheless, biosafety studies need to be addressed to implement live attenuated virus as real vaccine candidates [20].

In conclusion, this work helps to clarify the porcine immunological responses associated with cross-protection against ASFV, and provides the critical immune components that should

be targeted to achieve effective vaccination strategies. We demonstrate that a vaccine-dependent induction of an inflammatory response early after infection might play an important role in ASFV protection. While efforts to identify antigenic ASFV proteins to develop safer subunit vaccines are ongoing [11], this study represents a step forward to shift vaccine development from the current trial-and-error based approach to a more rational strategy.

## Materials and methods

### Ethics statement

Animal care and procedures were performed in accordance with the guidelines of the Good Experimental Practice and with the approval of the Ethics Committee on Animal Experimentation of the Generalitat de Catalunya (project codes: CEA-OH/10874/2 and CEA-OH/11241/1). All experiments were performed in the biosafety level 3 facilities at Centre de Recerca en Sanitat Animal (IRTA-CReSA, Barcelona).

### Viruses

BA71ΔCD2 is a LAV lacking the CD2v gene (EP402R), obtained by homologous recombination from the parental virulent BA71 ASFV strain [16]. BA71ΔCD2 was expanded in the established COS-1 cell line (ATCC). BA71ΔCD2 virus was titrated by immunoperoxidase monolayer assay (IPMA) as previously described. TCID50 titers obtained from IPMA were converted to pfu applying the Poisson distribution (TCID50/ml = 0.7 pfu/ml). The highly virulent Georgia2007/1 virus (genotype II) was kindly provided by Dr. Linda Dixon (WOAH reference laboratory, Pirbright Institute, UK). Georgia2007/1 ASFV was expanded in porcine alveolar macrophages (PAMs) obtained by lung lavage from healthy pigs. Georgia2007/1 virus stock, as well as BA71ΔCD2 and Georgia2007/1 titers in sera, were titrated by qPCR as previously described [16]. PRRSV ALL-183 strain and CSFV Margarita strain were kindly provided by Dr. Enric Mateu (UAB, Barcelona) and Dr. Llilianne Ganges (IRTA-CReSA, Barcelona), respectively.

### Animals and infections

Six- to eight-week-old Landrace X Large White male pigs were used. An acclimation period of seven days was allowed before vaccination. Animals were fed *ad libitum*. Results were obtained from four independent experiments. For experiment 1, animals were housed in three independent boxes (12 pigs/box), and each box was divided in two pens: one with six unvaccinated pigs (pen A), and another with six vaccinated pigs (pen B). Three different vaccine doses of the BA71ΔCD2 LAV [16] were tested (one in each box): low [$10^3$ plaque forming units (pfu)], intermediate ($3.3 \times 10^4$ pfu) and high ($10^6$ pfu). Each animal received 2 ml of the corresponding vaccine dose diluted in PBS via intranasal inoculation (1 ml/nostril). For each box, two pigs out of the six housed in the pen A were used as unvaccinated control animals that received PBS alone (6 unvaccinated control animals in total, 2 per box). The remaining four unvaccinated pigs/box in pen A were used for the direct-contact challenge. In detail, 20 days postvaccination (day -1 postchallenge), these animals were intramuscularly inoculated with $10^3$ gene equivalent copies (GEC) of the virulent Georgia2007/1 ASFV, and 24 hours after (day 0 postchallenge) vaccinated and unvaccinated pigs were challenged by direct contact with intramuscularly infected animals by mixing all the animals in each box (Fig 1A). Thus, A 2:1 ratio of vaccinated to intramuscularly challenged pigs was used. Intramuscularly inoculated pigs were sacrificed when evident ASF clinical signs were observed, which corresponded to 6 to 8 days after the challenge. The six unvaccinated controls pigs died between days 11 and 13 postchallenge, with no variation among the boxes. Pigs were bled and nasal and rectal swabs were

taken before and after vaccination (4, 7, and 14 days p.v.) and after challenge (0, 3, 6, 10, 13, and 20 days p.c.). For the other three experiments, pigs were vaccinated with $10^6$ pfu or inoculated with PBS as unvaccinated controls (following the same procedure described for experiment 1), and all animals were bled and euthanized at three weeks postvaccination. The number of animals used in each experiment is indicated in the results section and/or in the corresponding figure legends. Animal's health status was monitored according to a welfare schedule. Clinical signs were evaluated following standardized guidelines [75]. Briefly, animal behavior, body condition (prominence of vertebrae and ribs), presence of cyanosis, and digestive and respiratory signs were daily evaluated. Each parameter was scored from 0 to 3 according to the severity (0: normal; 1: mild; 2: moderate; 3: severe). Post-mortem examinations were carried out to confirm or discard the presence of ASF-compatible pathological lesions.

### Cell lines and primary cells

COS-1 cells were cultured at 37˚C in Dulbecco's modified Eagle medium (DMEM) supplemented with 5% heat-inactivated fetal calf serum (FCS), 50 μg/ml of gentamicin/ml (Sigma-Aldrich) and 2 mM L-glutamine (Invitrogen). PAMs were maintained in Roswell Park Memorial Institute (RPMI) 1640 medium (Gibco) supplemented with 10% heat-inactivated FCS (Cultek), 100 IU of penicillin/streptomycin/ml (Invitrogen), 2 mM L-glutamine (Invitrogen) and 0.5% nystatin. Peripheral blood mononuclear cells (PBMC) were separated from whole blood by density-gradient centrifugation with Histopaque 1077 (Sigma). Cellular suspensions of submandibular LN cells were obtained by incubating small pieces of tissue with a mix of 10 U/μl of DNaseI Recombinant, RNase-free (Roche) and 10 mg/ml of Collagenase type IV (Gibco Invitrogen) in RPMI 1640 medium (Gibco) at 37˚C for 30 min, gently mixing at 5 min intervals. Dissociated tissue was filtered through a 40 μm cell strainer, centrifuged and washed with PBS. Red blood cells from both PBMC and LN cell suspensions were lysed for 5 minutes with ammonium chloride. Final cell cultures were suspended in RPMI 1640 medium (Gibco) supplemented with 10% FCS, 100 IU of penicillin/streptomycin/ml (Invitrogen), 2 mM L-glutamine (Invitrogen) and 0.05 mM 2-mercaptoethanol. Trypan blue was used to assess cell viability. A detailed list of key reagents used is shown in the Key Resources Table.

### Quantitative PCR for the detection of ASFV

ASFV titers in sera and nasal swabs were assessed by SYBR Green qPCR targeting the ASFV PK gene as previously described [16]. Differential detection of BA71ΔCD2 was performed by probe-based qPCR targeting the LacI reporter gene only present in the genome of the BA71ΔCD2 vaccine virus. The primers and probe used were the following: LacI-Forward, 5'-TCGGTACCCTCGACGGATTT-3'; LacI-Reverse, 5'-CGCGGGAAACGGTCTGATAA-3'; LacI-Probe, 5'-VIC-CTAGATGAAACCAGTAACGTTATAC-MGBNFQ-3'. The qPCR recipe included 10 μL of Path-ID qPCR Master Mix (Applied Biosystems), 0.8 μL of 10 μM forward primer, 0.8 μL of 10 μM reverse primer, 0.4 μL of 10 μM probe, 2 μL of DNA extraction, and up to 20 μL of PCR-grade water. The program in the 7500 Fast Real-time PCR System (Applied Biosystems) was: 95˚C for 10 min; 40 cycles of 95˚C for 30 s, 60˚C for 1 min. A plasmid encoding the LacI gene was serially diluted and used as a standard template to determine the sensitivity of the qPCR. The detection limitation was 20 copies/reaction, with a Ct value of 34.93. Ct values below 34.93 were considered as positive.

### Enzyme-linked immunosorbent assay (ELISA)

ASFV-specific antibodies in pig sera were detected by the WOAH-approved ELISA based on soluble extracts from ASFV-infected cells [76]. The presence of positive sera was detected

using a peroxidase-conjugated anti-pig IgG at a 1/20,000 dilution (Sigma-Aldrich) as secondary antibody, and soluble 3,3′,5,5′-tetramethylbenzidine (TMB, Sigma-Aldrich) as specific peroxidase substrate. Reactions were stopped with 1 N $H_2SO_4$, and ELISA plates were read at a wavelength of 450 nm. Results were represented as the average absorbance [optical density (OD) values] of duplicates.

## IFNγ enzyme-linked immunosorbent spot (ELISpot) assay

IFNγ-secreting cells were assessed by ELISpot assay using purified mouse anti-pig IFNγ (clone P2G10, BD Pharmingen) as capture antibody and biotinylated mouse anti-porcine IFNγ antibody (clone P2C11, BD Pharmingen) as detection antibody, following a previously reported method [77]. Cells were stimulated with BA71ΔCD2 or Georgia2007/1 at a MOI of 0.2, and incubated for 16 hours at 37˚C, 5% CO2.

## Multiplex Luminex assay

PBMC were stimulated *in vitro* for 10 hours with BA71ΔCD2 at a MOI of 0.2, and cytokine levels were quantified in supernatants using the Luminex xMAP technology following the manufacturer's instructions. The measurements included IFNα, IFNγ, IL-1b, IL-10, IL-12p40, IL-4, IL-6, IL-8 and TNF (ProcartaPlex Porcine Cytokine & Chemokine Panel 1; Thermo-Fisher Scientific). Concentrations of each cytokine were calculated using the xPONENT software (Luminex). Since the sensitivity of multiplex Luminex assay is low, negative IFNγ results were further validated by ELISA (Kingfisher Biotech, Inc).

## Flow cytometry

For flow cytometric analysis, $10^6$ fresh PBMC obtained at day 0 p.c. were used per condition in U-bottom 96-well plates. Stimulations with BA71ΔCD2 were performed at a MOI of 0.2. For the detection of intracellular IFNγ and TNF expression, stimulation was performed for 6 hours plus 2 hours with Brefeldin A (BD GolgiPlug protein transport inhibitor) at 37˚C. For the detection of perforin expression, PBMC were stimulated for 48 hours. As negative and positive controls, RPMI and phorbol myristate acetate (PMA) plus ionomycin (at 5 ng/ml and 500 ng/ml, respectively) were used. Blockade of IFNγ was performed adding purified mouse anti-pig IFNγ (clone P2G10) at 5 μg/ml during the stimulation [78]. After stimulation cells were stained with LIVE/DEAD Fixable Violet Dead Cell Stain Kit or LIVE/DEAD Fixable Red Dead Cell Stain Kit (ThermoFisher Scientific) following manufacturer's instructions. Blockage of Fc receptors was performed with PBS 5% FCS for 15 min on ice prior to antibody staining. For extracellular staining, cells were incubated with the corresponding antibodies for 20 min on ice in FACS buffer (PBS 2% FCS). For intracellular staining, cells were fixed and permeabilized with the BD Cytofix/Cytoperm Kit (BD Biosciences) according to the manufacturer's protocol, and incubated with the corresponding antibodies for 30 min on ice in Perm/Wash buffer. The complete list of antibodies used is shown in the Key Resources Table. Samples were acquired on a BD FACSAria IIu flow cytometer (BD Biosciences) and data was analyzed using FlowJo v10.7.1 software (Tree Star Inc).

## RNA-seq library preparation and sequencing

PBMC were stimulated *in vitro* with BA71ΔCD2 or Georgia2007/1 at a MOI of 0.2 for 10 hours at 37˚C and kept at -80˚C in TRIzol Reagent (Invitrogen). Total RNA was isolated by phenol-chloroform method, and further purified using RNeasy MinElute spin columns (Qiagen) following the manufacturer's protocol. DNaseI (RNase-Free DNase Set, Qiagen)

treatment for 15 min at room temperature was performed to ensure RNA quality. Total RNA from PBMC was submitted for sequencing to the Genomics Unit of Centre for Genomic Regulation (CGR-CNAG). The quality and concentration of RNA were determined by an Agilent Bioanalyzer. The four samples per group with highest concentration of total RNA were selected for RNA-seq. Sequencing libraries were obtained after removing ribosomal RNA by a Ribo-Zero kit (Illumina). cDNA was synthesized and tagged by addition of barcoded Truseq adapters. Libraries were quantified using the KAPA Library Quantification Kit (KapaBiosystems) prior to amplification with Illumina's cBot. Four libraries were pooled and sequenced (single strand, 50 nts) on an Illumina HiSeq2000 sequencer to obtain 50–60 million reads per sample.

## Microfluidic quantitative PCR assay

Total RNA (110 ng) from PBMC (isolated as described in section 8) was reverse transcribed to cDNA using the PrimeScript RT reagent Kit (Takara, Japan, Cat. RR036A) following manufacturer's instructions. Primer design and validation were performed following previously described criteria [79]. The list of primers is provided in S8 Table. Gene expression levels were analyzed in duplicates using a microfluidic qPCR with the 96.96 Dynamic Array integrated fluidic circuit of the Biomark HD system (Fluidigm Corporation). Data was analyzed using the Fluidigm Real-Time PCR analysis software 4.1.3 and the DAG expression software 1.0.5.6 [80], and the relative standard curve method was applied (see Applied Biosystems user bulletin #2). Target gene expression levels were normalized against the average of three reference control genes (YWHAZ, RPL4 and GAPDH), and z-score normalized values were represented in a Heatmap.

## scRNA-seq library preparation and sequencing

Fresh LN cells obtained three weeks postvaccination (as described in *cell lines and primary cells* section) were stimulated with BA71ΔCD2 at a MOI of 0.2 for 16 hours at 37˚C. Cells were fixed with methanol as previously described [81], and kept at -80˚C until sent to the 10x Genomics scRNA-seq sequencing platform at CNAG-CRG (Barcelona, Spain). Methanol-fixed cells were rehydrated following the "Methanol Fixation of Cells for Single Cell RNA Sequencing" demonstrated protocol (10x Genomics). Briefly, fixed cells were equilibrated from -80˚C to 4˚C for 5 min and then centrifuged at 1,000 rcf for 5 min at 4˚C. Cell pellets were resuspended with Wash-Resuspension buffer, filtered through a 40 μm cell strainer (PluriSelect) and counted with the TC20 Automated Cell Counter (BioRad) to determine cell concentration. Cells were partitioned into gel bead-in-emulsions (GEMs) using the Chromium Controller system (10x Genomics) with the aim of a target cell recovery of 5,000 cells. Single-cell gene expression (GEX) libraries were prepared using the Chromium Single Cell 3′ Library & Gel Bead Kit v3.1 (10x Genomics) following manufacturer's instructions. In brief, after GEM–reverse transcription clean-up, cDNA was amplified using 13 cycles. cDNA quality control and quantification were performed using the Agilent Bioanalyzer High Sensitivity chip (Agilent Technologies). 10 to 50 ng of cDNA was used for library preparation and libraries were indexed by PCR using the Single Index Kit T Set A (10x Genomics). The size distribution and concentration of 3′ GEX libraries were verified using the Agilent Bioanalyzer High Sensitivity chip. Sequencing was performed using the Illumina NovaSeq6000 sequencer to obtain approximately 40,000 reads per cell.

## Bioinformatic analysis

**Bulk RNA-seq.** Illumina reads were mapped against Sus scrofa reference genome (Sscrofa11.1) using STAR software version 2.5.3a [82] with ENCODE parameters. Annotated

genes were quantified with RSEM version 1.3.0 [83] with default parameters using release 100 of Sus scrofa ENSEMBL annotation. Illumina reads were also mapped against the BA71 ASFV genome (accession code KP055815.1) using STAR software version 2.5.3a with ENCODE parameters. Annotated genes were quantified with RSEM version 1.3.0 with default parameters using KP055815.1 annotation. Differential expression analysis was performed with limma v3.42.3 R package [84], using TMM normalization. The voom function [85] was used to transform the count data into log2-counts per million (logCPM), estimate mean-variance relationship and to compute observation-level weights. These voom-transformed counts were used to fit the linear models. Given the paired nature of the data, the individual variation was blocked using the duplicateCorrelation function. Contrasts for pairwise comparisons were extracted, as well as contrasts for the interaction effect between treatment and vaccination status. Genes were considered DE if they had an adjusted p-value < 0.05. Functional enrichment analysis was performed using the DE genes with an absolute fold change (FC) > 1.5 using gprofiler2 v0.1.8 [86]. Sample similarities were inspected with a multidimensional scaling (MDS) plot using the top 500 most variable genes.

**scRNA-seq.** Sequencing reads were processed using CellRanger v.5.0.1 data [87], using the concatenated pig genome (Sscrofa11.1) and the ASFV strain BA71V assemblies (accession number KP055815) as a reference genome. The output folder "filtered_feature_bc_matrix" was used as input to perform downstream analysis with the R package Seurat 4.0.6 (R.4.1.2) [88]. To ensure good quality cells, only cell barcodes within the range of 200–2000 detected genes and < 5% mitochondrial content were kept for the analysis. Data was normalized with the SCTransform method and 3000 top most variable features were selected for integration with the FindIntegrationAnchors default of Seurat. UMAP was performed with the 30 first principal components, followed by the functions FindNeighbours and FindClusters with resolution 0.8 and 0.5. Cell cycle scoring and percentage of ASFV was also calculated per cell. Sub clustering of cluster 7, 8, 9 and 14 was performed with the function FindSubCluster and resolutions 0.2, 0.3, 0.5 and 0.3, respectively. Cell type annotation was performed manually looking for known immune pig marker genes. Differential expression between Unvaccinated and Vaccinated across each cell type was performed with the function FindMarkers with min.pct = 0.25 and logfc.threshold = 0.25. GO enrichment analyses were performed using gProfiler functional profiling (https://biit.cs.ut.ee/gprofiler/gost).

## Statistical analyses

Graphs were created and analyzed using Prism version 8.3.0. software (GraphPad). Statistical tests used are indicated on each figure legend. Statistical significance was set at p < 0.05 and is displayed in GraphPad style (p > 0.05 ns, * p ≤ 0.05, ** p ≤ 0.01, *** p ≤ 0.001, **** p ≤ 0.0001). A Fisher's exact test was performed for the comparison of transcript counts from Sus scrofa and ASFV obtained by RNA-seq from stimulated and unstimulated PBMC. Statistical tests for the microfluidic quantitative PCR dataset were performed in R v4.1.2 (S5 Table); the within-group comparisons (unvaccinated and vaccinated) for the expression values of the various treatments were performed using a paired pairwise t-test, whereas the between-group comparisons were done with an independent pairwise t-test.

## Supporting information

**S1 Fig. BA71ΔCD2-vaccinated pigs do not show major clinical signs.** (A) Rectal temperatures from individual animals in each group. (B) Clinical scores measured throughout the experiment. Each row represents an animal within the group.
(DOCX)

**S2 Fig. BA71ΔCD2-vaccinated pigs protected against Georgia2007/1 direct-contact challenge do not show major clinical signs.** A) Rectal temperatures from individual animals in each group. (B) Clinical scores measured throughout the experiment. Each row represents an animal within the group.
(DOCX)

**S3 Fig. BA71ΔCD2-vaccinated pigs protected against Georgia2007/1 direct-contact challenge show low virus levels in sera and nasal cavities and high ASFV-specific antibody titers.** Virus titers in (A) sera and (B) nasal swabs measured by qPCR at the indicated time points after Georgia2007/1 direct-contact challenge. (C) ASFV-specific antibody levels in sera from vaccinated and unvaccinated pigs assessed by ELISA at the indicated time points.
(DOCX)

**S4 Fig. *In vitro* ASFV-specific stimulation induces robust transcriptomic changes in PBMC from BA71ΔCD2-immunized pigs including both adaptive and innate transcriptomic signatures.** (A) Multidimensional scaling analysis of the normalized RNA-seq expression levels (log2CPM). (B) Venn diagram showing the number of overlapping and unique DE genes identified in the unvaccinated and vaccinated groups. (C) List of representative GO terms enriched in DE genes from BA71ΔCD2-stimulated PBMC from unvaccinated and vaccinated pigs. The size of the dots represents the number of DE genes associated with the GO term, and the color indicates the negative log10 value of the false discovery rate (FDR). (D) Heatmap depicting normalized RNA-seq-derived log2CPM values of representative DE genes (extension of Fig 2C).
(DOCX)

**S5 Fig. Pigs vaccinated with $10^6$ pfu of BA71ΔCD2 from which submandibular lymph node cells were used do not show major clinical signs.** (A) Rectal temperatures from individual animals in each group. (B) Clinical scores measured throughout the experiment. Each row represents an animal within the group.
(DOCX)

**S6 Fig. IFNγ ELISpot and analysis of scRNA-seq data in submandibular LN cells.** (A) Pigs were vaccinated with $10^6$ pfu of BA71ΔCD2 (n = 6) and three weeks later levels of ASFV-specific cells in submandibular LN were measured by IFNγ ELISpot using BA71ΔCD2 as stimulus. Unvaccinated pigs (n = 6) were used as negative control. (B) Number of genes differentially expressed between the unvaccinated and the vaccinated pig in each cluster. (C) Number of cells in each cluster identified by scRNA-seq. (D) Pearson's correlation between the number of DE genes and the number of cells in each cluster. Each dot represents a cluster, and the value of the total number of cells is the result of the addition of cells from each sample.
(DOCX)

**S7 Fig. Gene ontology enrichment analysis of scRNA-seq-derived clusters shows the activation of innate immunity in cells from the vaccinated pig.** List of representative GO terms enriched for each cluster in DE genes from BA71ΔCD2-stimulated submandibular LN cells from the vaccinated pig. The size of the dots represents the number of DE genes associated with the GO term, and the color indicates the negative log10 value of the false discovery rate (FDR).
(DOCX)

**S8 Fig. Violin plots depicting scRNA-seq-derived expression levels of representative ISG in each cluster.** Asterisks denote differential expression: ** p value adjusted ≤ 0.01, *** p value

adjusted $\leq 0.001$.
(DOCX)

**S9 Fig. A cytotoxic recall response in BA71ΔCD2-vaccinated animals is revealed by scRNA-seq analysis of LN cells but not by RNA-seq analysis of PBMC stimulated for a shorter period.** (A) Violin plots showing expression levels from scRNA-seq-derived data of *CCR7*, *CXCR4* and profilin 1 (*PFN1*) in CTLs from LN cells after 16 hours of *in vitro* ASFV-specific stimulation. (B) RNA-seq-derived expression levels as log2CPM values of representative cytotoxic markers in PBMC after 10 hours of *in vitro* ASFV-specific stimulation.
(DOCX)

**S1 Table. Pigs vaccinated with BA71ΔCD2 do not show viral DNA neither in serum nor in whole blood.** The table shows genomic equivalent copies (GEC)/ml assessed by qPCR assay targeting the ASFV PK gene. Samples obtained at day 0, 14 and 21 postvaccination from pigs receiving $10^6$ pfu of BA71ΔCD2 were used. Undet: Undetectable.
(XLSX)

**S2 Table. BA71ΔCD2-vaccinated animals do not show BA71ΔCD2 replication in sera after Georgia2007/1 challenge.** The table shows Ct values obtained using Taqman primer/probe sets targeting LacI, only present in the BA71ΔCD2 genome. Serum samples from BA71ΔCD2-vaccinated pigs at days 10 and 13 after Georgia2007/1 challenge were tested. Positive controls of the assay were: 1) nasal swabs samples from BA71ΔCD2-vaccinated pigs at days 7 and 14 postvaccination, and 2) serum from a naïve pig where BA71ΔCD2 was added.
(XLSX)

**S3 Table. GO terms enriched in RNA-seq-derived DE genes from PBMC from unvaccinated animals stimulated with BA71ΔCD2.**
(XLSX)

**S4 Table. GO terms enriched in RNA-seq-derived DE genes from PBMC from vaccinated animals stimulated with BA71ΔCD2.**
(XLSX)

**S5 Table. Statistical analysis results from microfluidic quantitative PCR assay data.**
(XLSX)

**S6 Table. BA71ΔCD2-vaccinated animals show low or undetectable levels of BA71ΔCD2 in submandibular lymph node cells.** Submandibular lymph node cells from control or BA71ΔCD2-vaccinated ($10^6$ pfu) pigs were obtained three weeks after vaccination. Cells were left untreated (Mock) or stimulated *in vitro* for 16 hours with BA71ΔCD2 (positive control) prior DNA extraction and qPCR for the detection of ASFV. Results are shown as ASFV GEC/$10^6$ cells. Samples marked with an asterisk were used in the scRNA-seq analysis. Undet: Undetectable.
(XLSX)

**S7 Table. Statistical analysis results from Fisher's exact test comparing cell proportions of scRNA-seq clusters between samples.**
(XLSX)

**S8 Table. List of primers used for microfluidic quantitative PCR assay.**
(XLSX)

**S9 Table. Key Resources Table.**
(DOCX)

## Acknowledgments

We thank Anna Barceló, Jordi Rodon, Álvaro López, and the Animal Facility unit from IRTA-CReSA for their excellent technical support; CNAG-CRG for assistance with single cell RNA-seq; Dr. Carmina Gallardo for providing ASFV cell lysate for ASFV-specific ELISA; Dr. Javier Domínguez and Dr. Ángel Ezquerra for providing antibodies for flow cytometry; and Dr. Enric Mateu and Dr. Llilianne Ganges for providing PRRSV and CSFV virus stocks, respectively. We acknowledge support of Red de Investigación en Sanidad Animal (RISA) and World Organisation for Animal Health (WOAH). We also acknowledge support of the Spanish Ministry of Science and Innovation through the EMBL partnership and the Instituto de Salud Carlos III; the Centro de Excelencia Severo Ochoa; the Generalitat de Catalunya through Departament de Salut and Departament d'Empresa i Coneixement and CERCA Programme; and the European Regional Development Fund by the Spanish Ministry of Science and Innovation corresponding to the Programa Operativo FEDER Plurirregional de España (POPE) 2014–2020 and by the Secretaria d'Universitats i Recerca, Departament d'Empresa i Coneixement of the Generalitat de Catalunya corresponding to the Programa Operatiu FEDER de Catalunya 2014–2020.

## Author Contributions

**Conceptualization:** Fernando Rodríguez, Jordi Argilaguet.

**Data curation:** Anna Esteve-Codina, Jordi Argilaguet.

**Formal analysis:** Laia Bosch-Camós, Anna Esteve-Codina, Beatriz Martín-Mur.

**Funding acquisition:** Lihong Liu, Boris Gavrilov.

**Investigation:** Laia Bosch-Camós, Uxía Alonso, Chia-Yu Chang, Beatriz Martín-Mur, Marta Muñoz, María J. Navas, Marc Dabad, Enric Vidal, Sonia Pina-Pedrero, Patricia Pleguezuelos, Ginevra Caratù, Stanimira Bataklieva.

**Methodology:** Anna Esteve-Codina, María L. Salas, Fernando Rodríguez, Jordi Argilaguet.

**Project administration:** Francesc Accensi, Fernando Rodríguez, Jordi Argilaguet.

**Resources:** Boris Gavrilov, Fernando Rodríguez, Jordi Argilaguet.

**Software:** Anna Esteve-Codina, Beatriz Martín-Mur.

**Supervision:** Fernando Rodríguez, Jordi Argilaguet.

**Validation:** Jordi Argilaguet.

**Visualization:** Laia Bosch-Camós, Uxía Alonso, Anna Esteve-Codina, Jordi Argilaguet.

**Writing – original draft:** Laia Bosch-Camós, Uxía Alonso.

**Writing – review & editing:** Fernando Rodríguez, Jordi Argilaguet.

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
