## [Decision Letter · Decision Letter 0]

30 Aug 2022

Dear Dr. Argilaguet,

Thank you very much for submitting your manuscript "Cross-protection against African swine fever virus upon intranasal vaccination is associated with an adaptive-innate immune crosstalk" for consideration at PLOS Pathogens. As with all papers reviewed by the journal, your manuscript was reviewed by members of the editorial board and by several independent reviewers. The reviewers appreciated the attention to an important topic. Based on the reviews, we are likely to accept this manuscript for publication, providing that you modify the manuscript according to the review recommendations.

Reviews received from two reviewers agree that your manuscript provides important novel information relevant for understanding correlates of protection following immunization of pigs with a previously characterized live attenuated African swine fever virus strain BA71ΔCD2 and heterologous challenge with a virulent genotype II strain. However both raise important points and caveats that you will need to respond to and modify the manuscript as required prior to acceptance. Please respond point by point indicating the modifications that have been introduced.

Sincerely,

Linda Kathleen Dixon

Associate Editor

PLOS Pathogens

Klaus Früh

Section Editor

PLOS Pathogens

Kasturi Haldar

Editor-in-Chief

PLOS Pathogens

orcid.org/0000-0001-5065-158X

Michael Malim

Editor-in-Chief

PLOS Pathogens

orcid.org/0000-0002-7699-2064

Reviews received from two reviewers agree that your manuscript provides important novel information relevant for understanding correlates of protection following immunization of pigs with a previously characterized live attenuated African swine fever virus strain BA71ΔCD2 and heterologous challenge with a virulent genotype II strain. However both raise important points and caveats that you will need to respond to and modify the manuscript as required prior to acceptance. Please respond point by point indicating the modifications that have been introduced.

Reviewer Comments (if any, and for reference):

Reviewer's Responses to Questions

**Part I - Summary**

Reviewer #1: Bosch-Camos et al present here an analysis of the local and systemic immune response triggered by intranasal vaccination with attenuated African swine fever virus (ASFV). ASFV is a very important virus, expanding in a current zoonosis with a strong impact worldly on the porcine industry. Herein, the authors carried out four independent in vivo experiments on pigs and used up to date technics such as RNAseq and single cell RNAseq technics, backed up by flow cytometry analysis, allowing an accurate and robust description of the immune cells involved upon ASFV protective immune setting new references for further studies.

Stand against these strong premises, it is a pity that the authors don’t go a little deeper on one of their main findings, the importance of myeloid cells activation or training in the efficacy of the BA71ΔCD2.

In conclusion, although this work can be improved by deepening some investigations, it deserves publication in Plos Pathogens.

Reviewer #2: In this manuscript, the authors investigated the immune response of domestic pigs to an infection challenge with African swine fever virus following vaccination with the live genotype I attenuated deletion mutant BA71ΔCD2. Pigs inoculated with a high dose (10^6) of BA71ΔCD2 survived a challenge when placed in contact with animals infected with the virulent heterologous genotype II strain Georgia 2007. Pigs that received lower doses (10^4.5 and 10^3) showed moderate to little protection against challenge in a dose-dependent manner, whereas the unvaccinated group succumbed 11-13 days post contact.

Protection appeared to correlate somewhat with ASFV-specific antibodies in serum and number of IFNgamma+ PBMCs prior to contact challenge. Bulk RNAseq analysis of PBMCs of vaccinated animals revealed that restimulation induced strong innate and adaptive transcript profiles upon re-stimulation in vitro. In a second vaccination study (single high dose), PBMCs were re-stimulated with a homologous and heterologous strain in vitro 3 weeks after vaccination and a subset of transcripts were re-tested by digital PCR. Finally, submandibular lymph nodes from one vaccinated and one control pig were subjected to single cell RNA-seq 3 weeks after nasal vaccination revealing specific changes in lymph node cell subsets and associated transcriptional responses.

The study provides new insights that will be informative for understanding correlates of protection that may help in rational design of vaccines against ASFV, arguably the pig pathogen of the greatest importance for animal health. The manuscript is well written, and the data accurately presented.

**Part II – Major Issues: Key Experiments Required for Acceptance**

Reviewer #1: Major remarks:

- Line 180, line 403, the authors evoke the possibility of monocytes training. Trained immunity has been shown to be induced by attenuated/recombinant vaccines (for review Goodridge Nat Rev Immunol 2016). One of the best publication, close to the work presented here, is Yao Cell 2018. An analysis of the alveolar macrophages phenotype and function would be out of the scope here, however one experiment investigating the in vitro cytokinic responses of post-vaccination monocytes to some TLR ligands such as LPS, Poly IC, R848, would add some mechanistic as well as a very interesting opening for further studies, in this highly descriptive work. In all cases some sentences in the discussion on the possibility of an indirect monocyte training, as observed in Yao et al. would be helpful.

- In the same line, the authors might restimulate PBMC with overlapping peptides of AFSV main antigens in order to untangle the innate immune cells activation trigger by the whole virus, from the antigen specific response triggered by the T lymphocytes.

- Line 268 and Fig 4B, please depict a really quantitative figure (mean+/- SD?) and not just this figure leading to a fuzzy visual feeling.

- Line 279: Figure 4F raised a very important point, as stressed by the authors in the discussion. However, no significant marks are depicted. If these data are not significant, either try to plot the % of reduction of TNF producing cells to get significant differences or make new experiments. If this is not possible, please don’t claim that ‘TNF production was abolished by blocking of IFNg’ (Line 279), and use more caution line 405 when stating ‘the dramatic reduction of TNF-producing macrophages when blocking IFNg’.

- Line 348: It is quite disturbing to validate LN scRNAseq using FACS analysis of PBMC. Please justify.

Reviewer #2: Specific comments that need to be addressed are:

1- One caveat to be discussed is that the vaccinated pigs are challenged only 3 weeks after vaccination. The vaccine virus is still detectable in serum and swabs of vaccinated animal (S2A/B Fig, T=0). It is likely that more virus will be found in whole blood (usually 2-3 logs higher than in serum) and potentially also in the submandibular lymph nodes. What is the impact of the presence of the virus on the innate response during in vitro recall and in vivo challenge?

2- Did the author try a genotype-specific qPCR to detect any sign of replication of the vaccine strain upon challenge with Georgia 2007 in the first experiment?

3- What is the status of viremia and virus load in the lymph node in the second experiment at the time of recall and RNA seq. Was it tested?

4- Did the authors record clinical signs and temperature after vaccination? It should be shown in supplementary materials for both experiments if available.

5- The adaptive-immune crosstalk link is somewhat speculative (line 387-389; 404-406). Blockade of IFNg blocked TNF expression but the mechanistic (“dependence”) link to CXCL10-mediated inflammatory response was not shown, only association can be deduced from RNAseq data.

6- The lack of IFNg detection by ELISA is puzzling in Figure 1E given the ELISPOT and transcriptomic results. Can the authors exclude a technical issue with the multiplex ELISA using a positive control stimulation of naïve PBMCs? (we had issues with very low bead IFNgamma bead counts with the same kit).

7- On the same point, the proposed justification as to why there is no IFNgamma in supernatants “the presence of a low number of ASFV-specific memory T cells in PBMC” (lines 178-179) does not make sense. The authors also contradict this point lines 196/197.

8- In figure 4E, the specificity of the myeloid response in producing TNF is not clearly demonstrated. It would be better shown by adding a non-specific stimulation of PBMC with the heterologous ASFV Georgia 2007 virus or better with another type of virus infecting porcine macrophages. This may help reveal whether the myeloid response is truly specific or whether there is just non-specific innate priming (e.g. trained immunity response, epigenetic changes) that would not be as long lasting. The blockade of IFNg, while elegant and informative, is not sufficient to rule out an innate source of the cytokine (e.g., NK cells).

**Part III – Minor Issues: Editorial and Data Presentation Modifications**

Reviewer #1: Minor remarks

- Figures’ legends are included in the main text.

-In the abstract and/or author summary, it would be good to know a little more on the possibility to use this vaccine in the field, please evoke the biosafety.

- Can you add some sentences on the mechanisms involved in the attenuation through CD2v depletion? For instance, do the attenuated virus still infect monocytes?

- Line 100, please specify that the memory CD4 and CD8 expanding after in vitro restimulation came from infected/vaccinated animals.

- Line 146, Figure 1D, the authors might process to a principal component analysis including all the read out of the vaccination.

- Line 177, the sentence is not clear: ELISpot result does not ‘suggest the presence of a low number of ASFC-specific memory T cells in PBMC’, it reveals and precisely measure this low number.

- Line 183: Please specify how you assure yourself of the random selection? Also specify here that you chose the highest dose of vaccine in the following experiments.

- Legend Fig2A and 2B, please specify: left unstimulated, right ASFV stimulated.

Reviewer #2: Minor comments

9- The setup of the first experiment needs to be clarified. Line129, the group of unvaccinated animals is not initially presented. Is it distinct from the challenged contact animals?

10- Methods indicate that a ratio of 2:1 of vaccinated to intramuscularly injected pigs. However, it is not clear if all animals in the 4 groups (3 vaccinated and control) and the contact infected animals were all in a single large stable/room or if there were subgroups. If there were subgroups, please describe the setup. The heterogeneity of the disease course in the 2 groups receiving low dose of vaccines may be random or caused the variability of viral shedding by the contact intramuscularly infected pigs.

11- What does “GP style” mean in figure legends?

12- Figures 2 and 3. XCL1 is a chemokine. It is grouped with cytokines.

13- S2 Fig. y axis labels and titles are missing.

14- The use of cDC_1 and cDC_2 as a labels for the subsets of cross-presenting DCs is not judicious as this may be confused with usual acronym for conventional dendritic cells (cDC1 and cDC2). Perhaps xDC may be a better choice to avoid confusing readers.

15- Line 342, spell out profilin 1 (to avoid mix up with perforin).

16- Line 351: “significantly” should be used only in relation to statistical analysis.

17- L384, the claim of priority (“first”) is perhaps unnecessary.

18- Fig 5A and 6A, the colors of some subsets are hard to decipher. The paucity of gd T cells is surprising. Have the authors confirmed presence T cell receptor alpha or beta chains in other T cell subsets?

PLOS authors have the option to publish the peer review history of their article (what does this mean?). If published, this will include your full peer review and any attached files.

Reviewer #1: No

Reviewer #2: **Yes: **Charaf Benarafa

Figure Files:

Data Requirements:

Reproducibility:

References:

---

## [Decision Letter · Decision Letter 1]

17 Oct 2022

Dear Dr. Argilaguet,

We are pleased to inform you that your manuscript 'Cross-protection against African swine fever virus upon intranasal vaccination is associated with an adaptive-innate immune crosstalk' has been provisionally accepted for publication in PLOS Pathogens.

Best regards,

Linda Kathleen Dixon

Associate Editor

PLOS Pathogens

Klaus Früh

Section Editor

PLOS Pathogens

Kasturi Haldar

Editor-in-Chief

PLOS Pathogens

orcid.org/0000-0001-5065-158X

Michael Malim

Editor-in-Chief

PLOS Pathogens

orcid.org/0000-0002-7699-2064

Thank you for the revision of your manuscript and response to reviewers' points raised. The manuscript can now be accepted although we request two additional minor changes for clarity. The first is raised by Reviewer 1 in minor issues, "I would just ask the authors to specify that the results evoked lines 280-282 are not shown (data not shown)". The second request is to specify more clearly which ASFV gene was the target for qPCR by providing the gene assignation. In the text (eg line 603 and Table S2) the gene is designated only with PK. Thank you for submitting your manuscript to PLOS Pathogens

Reviewer Comments (if any, and for reference):

Reviewer's Responses to Questions

**Part I - Summary**

Reviewer #1: The authors responded to their best to my queries.

**Part II – Major Issues: Key Experiments Required for Acceptance**

Reviewer #1: (No Response)

**Part III – Minor Issues: Editorial and Data Presentation Modifications**

Reviewer #1: For clarity, I would just ask the authors to specify that the results evoked lines 280-282 are not shown (data not shown).

PLOS authors have the option to publish the peer review history of their article (what does this mean?). If published, this will include your full peer review and any attached files.

Reviewer #1: **Yes: **Nicolas BERTHO

---

## [Editor Report · Acceptance letter]

26 Oct 2022

Dear Dr. Argilaguet,

We are delighted to inform you that your manuscript, "Cross-protection against African swine fever virus upon intranasal vaccination is associated with an adaptive-innate immune crosstalk," has been formally accepted for publication in PLOS Pathogens.

Best regards,

Kasturi Haldar

Editor-in-Chief

PLOS Pathogens

orcid.org/0000-0001-5065-158X

Michael Malim

Editor-in-Chief

PLOS Pathogens

orcid.org/0000-0002-7699-2064